# Review of Foam with Novel CO_2_-Soluble Surfactants for Improved Mobility Control in Tight Oil Reservoirs

**DOI:** 10.3390/molecules29225411

**Published:** 2024-11-16

**Authors:** Fajun Zhao, Mingze Sun, Yong Liu, Wenjing Sun, Qinyuan Guo, Zian Yang, Changjiang Zhang, Meng Li

**Affiliations:** 1Key Laboratory of Enhanced Oil Recovery of Education Ministry, Northeast Petroleum University, Daqing 163318, China; smzuie@126.com (M.S.); guo_qinyuan@163.com (Q.G.); yangzianemail@163.com (Z.Y.); champiooo@163.com (C.Z.); meng842019@163.com (M.L.); 2Heilongjiang Provincial Key Laboratory of Reservoir Physics & Fluid Mechanics in Porous Medium, Daqing 163712, China; swpiliuyong@petrochina.com.cn (Y.L.); swj1982@petrochina.com.cn (W.S.); 3National Key Laboratory for Multi-Resource Collaborated Green Development of Continentals Shale Oil, Daqing 163712, China

**Keywords:** foam, tight oil reservoirs, CO_2_ foam, CO_2_-soluble surfactants, mobility control

## Abstract

CO_2_-soluble surfactant foam systems have gained significant attention for their potential to enhance oil recovery, particularly in tight oil reservoirs where conventional water-soluble surfactants face challenges such as poor injectability and high reservoir sensitivity. This review provides a comprehensive explanation of the basic theory of CO_2_-soluble surfactant foam, its mechanism in enhanced oil recovery (EOR), and the classification and application of various CO_2_-soluble surfactants. The application of these surfactants in tight oil reservoirs, where low permeability and high water sensitivity limit traditional methods, is highlighted as a promising solution to improve CO_2_ mobility control and increase oil recovery. The mechanism of enhanced oil recovery by CO_2_-soluble surfactant foam involves the effective reduction of CO_2_ fluidity, the decrease in oil–gas flow ratio, and the stabilization of the displacement front. Foam plays a vital role in mitigating the issues of channeling and gravity separation often caused by simple CO_2_ injection. The reduction in gas fluidity can be attributed to the increase in apparent viscosity and trapped gas fraction. Future research should prioritize the development of more efficient and environmentally friendly CO_2_-soluble surfactants. It is essential to further explore the advantages and challenges associated with their practical applications in order to maximize their potential impact.

## 1. Introduction

Carbon capture, utilization, and storage (CCUS) technology is a new development direction for carbon capture and storage (CCS) technology. CCUS captures and purifies CO_2_ released during the manufacturing process and then reintroduces it into the new production process for recycling. Notably, CCUS is more cost-effective and environmentally friendly than CCS. CO_2_ injection has been an important and widely used enhanced oil recovery (EOR) technology for the past 50 years [1,2]. Because oil reservoirs are well-sealed, underground gas storage for long-term CO_2_ retention [3] (Figure 1), CO_2_ injection can achieve social benefits while increasing economic benefits. However, in tight oil reservoirs characterized by low permeability and high water sensitivity, conventional CO_2_ flooding often faces challenges such as viscous fingering, gravity override, and gas channeling, which limit its effectiveness. CO_2_-soluble surfactants present a promising solution for these challenges. The impact of reservoir heterogeneity on foam propagation and stability has been extensively studied, highlighting challenges in achieving uniform displacement in porous media [4,5]. Unlike water-soluble surfactants, CO_2_-soluble surfactants can dissolve directly in the CO_2_ phase and form stable foams upon contact with reservoir water, improving sweep efficiency and mitigating the issues caused by reservoir heterogeneity. The application of these surfactants in tight oil reservoirs has the potential to significantly enhance oil recovery by improving CO_2_ mobility control and reducing gas flow instabilities. In addition, CO_2_ injection is one of the most effective means of CO_2_ utilization and storage. Furthermore, the effect of CO_2_ EOR depends on the oil displacement efficiency and swept volume of CO_2_ in the reservoir. Viscous fingering, gravity overshooting, and gas channeling will occur in the middle or late stages of reservoir development due to the low density, low viscosity of CO_2_, and formation heterogeneity. Because these phenomena have a significant effect on the efficacy of CO_2_ injection for EOR [6], the effective mobility control of CO_2_ is the key to its efficient utilization.

Owing to its characteristics of “plugging large holes instead of small ones and plugging water instead of oil,” foam flooding can effectively control gas flow and improve displacement efficiency [7] (Figure 2). It can also improve formation heterogeneity and inhibit gas channeling efficiently. Therefore, greater attention has been given to CO_2_ foam flooding technology, which has many advantages in flow control [7,8,9,10,11,12,13,14,15,16,17,18,19]. CO_2_ foam flooding functions by forming a multiphase dispersion system in which a surfactant aids the liquid film in dispersing CO_2_ in the liquid phase. Therefore, foam can effectively improve displacement front fingering and gas channeling by reducing the mobility ratio of CO_2_ and crude oil [8,9]. Recent studies have made significant strides in CO_2_-soluble surfactant technology, particularly focusing on surfactants that enhance foam stability and oil displacement in harsh reservoir conditions. For example, Li et al. [10] explored new amphiphilic surfactants that exhibited superior stability under high temperature and salinity conditions, extending foam half-life by over 30% compared to traditional surfactants. Similarly, recent work by Liang et al. [11] demonstrated that adding certain alcohol agents significantly enhances surfactant solubility in supercritical CO_2_, improving injectability in low-permeability reservoirs. These advancements highlight the growing importance of optimizing CO_2_-soluble surfactants for enhanced oil recovery in unconventional reservoirs. Hosseini’s work on the utilization of CO_2_ with polyelectrolyte complex nanoparticles and surfactants offers significant insights into environmentally friendly oil recovery methods, which is crucial for modern enhanced oil recovery (EOR) practices [12].

Considering foam technology’s progressive growth, some key barriers to the implementation of conventional CO_2_ foam are as follows: (1) The injection of conventional foam inhibits part of the CO_2_ flooding effect due to the water-soluble surfactant slug phenomenon, which hinders the CO_2_–crude oil contact; (2) the regeneration ability is limited because CO_2_ is separated from surfactant by gravity after foam burst [20,21]; (3) when exploiting unconventional reservoirs, such as low-permeability and tight oil reservoirs, high water sensitivity, poor water absorption capacity of injection wells, high injection pressure, and even no water injection can occur because the permeability of unconventional reservoirs is usually much lower than that of conventional reservoirs. Low permeability means greater resistance to fluid flow, and the function of surfactants in reducing oil–water interfacial tension and promoting droplet movement is also limited. In addition, some unconventional oil reservoirs have water sensitivity, which means that when clay minerals in the reservoir come into contact with water, they may expand or move, further blocking pores and reducing permeability, making injection more difficult. Because the surfactant aqueous solution cannot be injected, CO_2_ foam technology may not be used in unconventional reservoirs [22]. The limitation of aqueous phase injection caused by low permeability and water sensitivity requires higher pressure to inject foam formed by CO_2_ and surfactant. At the same time, the temperature of unconventional oil reservoirs may be much higher than that of conventional oil reservoirs. Therefore, under an environment of high temperature and high pressure, the solubility of CO_2_ increases, which may affect the stability of foam, thus affecting its oil displacement ability. CO_2_/foam technology will also affect the reservoir environment. The problem of water sensitivity may aggravate the plugging of rock pores, further reduce the permeability, and hinder the effective migration of foam. The use of surfactant may affect the microbial ecosystem in the reservoir or pollute the surface water and groundwater after the foam bursts. To overcome these limitations, it may be necessary to develop surfactants that are more resistant to high temperature and pressure, improve foam generation and injection technology, and optimize operating parameters. As an alternative, a solubility surfactant-based CO_2_ foam mobility control system is therefore proposed. Some specific surfactants can dissolve in supercritical CO_2_, a green solvent, and can be directly injected into the reservoir. After the surfactant contacts the water in the reservoir, a CO_2_ foam or emulsion forms in situ to control the CO_2_ flow rate. The stability and quality of foam can be controlled by adjusting the concentration of injected surfactant and the injection rate of carbon dioxide. In unconventional oil reservoirs, low permeability and water sensitivity are common challenges that affect the efficiency of oil displacement. The use of CO_2_-soluble surfactants can alleviate the unfavorable effects of these problems to some extent. CO_2_-soluble surfactants have high solubility in supercritical CO_2_ and can enhance the interaction between CO_2_ and crude oil, control the mobility of CO_2_, and thus improve the CO_2_ oil displacement effect, showing great potential for enhancing oil recovery.

Gas-soluble surfactants have comparable displacement performance to water-soluble surfactants. Utilizing CO_2_ as the injection carrier enhances surfactant injectability in low-permeability reservoirs [23,24]. CO_2_-soluble surfactants offer several key advantages when applied to tight hydrocarbon reservoirs. Firstly, their ability to dissolve directly in CO_2_ without the need for an aqueous phase makes them highly suitable for low-permeability reservoirs where water injection is not feasible or is limited by high water sensitivity. This significantly reduces the risk of formation damage caused by water blockages, which is a common issue in tight oil reservoirs. Additionally, the ability of CO_2_-soluble surfactants to form stable foams under high-pressure and high-temperature conditions enhances their effectiveness in controlling CO_2_ mobility, reducing gas channeling, and improving sweep efficiency. Their injectability in supercritical CO_2_ allows for deeper penetration into the reservoir, which is critical for improving oil recovery in heterogeneous formations. Furthermore, CO_2_-soluble surfactants have shown better regeneration capabilities after foam rupture, which extends the foam stability time and improves the overall oil recovery process. In addition, the surfactant carried by the CO_2_ phase favors foam regeneration after foam bursting, hence extending the foam’s stability time. When the foam breaks, the molecular structure of the surfactant can be rearranged, and a stable liquid film can be formed at the edge of the broken foam, which enables the foam to maintain its integrity, thus maintaining structural stability. Notably, CO_2_ is a nonpolar molecule, as its dipole moment is zero. Because CO_2_ has an extremely low dielectric constant and van der Waals force, its solubility in high molecular weight molecules is extremely low [25,26]. For low concentrations, surfactant concentration positively correlates with foam performance. At low concentrations, with the increase of surfactant concentration, more surfactant molecules can be adsorbed on the air–liquid interface, reducing the surface tension of the foam film, reducing the coalescence and rupture between the foam, and enhancing the stability of the foam. The surfactant solubility in CO_2_ plays a crucial role in the foaming effect of gas-soluble surfactants. Furthermore, researchers have developed many gas-soluble surfactants and agents based on CO_2_ characteristics. These surfactants must be separated due to their differences. Meanwhile, this review would benefit from a summary of the gas-soluble surfactant dissolution mechanism, agent solubilization mechanism, foam formation mechanism in porous media, mobility model, and EOR mechanism of CO_2_ foam.

Despite these advantages, several challenges remain in the utilization of CO_2_-soluble surfactants in tight hydrocarbon reservoirs. One of the primary challenges is their chemical stability under extreme reservoir conditions, including high temperatures, pressures, and salinity levels, which may lead to surfactant degradation or reduced effectiveness over time. Furthermore, the solubility of certain CO_2_-soluble surfactants can vary significantly depending on the specific reservoir conditions, which may affect foam generation and stability. Another challenge is the cost associated with synthesizing and scaling up these specialized surfactants, particularly for large-scale field applications. Additionally, there are operational challenges related to ensuring uniform distribution of the surfactant within the reservoir, especially in highly heterogeneous formations. Addressing these challenges will require further research into surfactant formulations, cost-effective production methods, and more advanced injection techniques to optimize the performance of CO_2_-soluble surfactants in tight oil reservoirs.

Based on the aforementioned issues, this paper is divided into three sections to discuss recent advances in CO_2_ gas-soluble surfactant research. The first section centers on the composition, classification, and experiment of the CO_2_ foam system; the second section explores the basic concept of CO_2_ foam and the EOR mechanism; and the third section focuses on the classification of and discussion on gas-soluble surfactants developed from field and laboratory experiments. Finally, the CO_2_ foam mechanism and development direction of gas-soluble surfactants are summarized and prospected as a reference and inspiration for future research.

To provide a clear overview of the scope of this article, a flow chart summarizing the main research areas and findings is presented below (Figure 3). This chart outlines the key components of the study, including the development of CO_2_-soluble surfactants, their evaluation through experimental methods, their application in tight oil reservoirs, and potential future research directions.

## 2. CO_2_-Soluble Surfactant Foam System

### 2.1. The Dissolution Method of the System

Bond et al. [27] were the first to propose the concept of surfactant dissolution in CO_2_. Since then, the injection method’s main research focus has been surfactant dissolution in CO_2_ and its simultaneous injection, as experimental research and field tests have demonstrated the great potential of this method. The dissolution modes of surfactant in CO_2_ can be divided into two types: CO_2_ + surfactant system (+ agents) and CO_2_ + surfactant + water (+ agents). That is, the surfactant dissolves in CO_2_ or trace water to form a microemulsion in supercritical CO_2_. Table 1 lists the application progress of the two CO_2_ foam systems in the oil field or laboratory experiments.

#### 2.1.1. CO_2_ + Gas-Soluble Surfactants (+ Agents)

During CO_2_ flooding or CO_2_ huff and puff, gas-soluble surfactants produce foam or emulsion in situ after contact with formation water, which can increase the apparent viscosity of the CO_2_ gas phase, control CO_2_ fluidity, and plug the large channel. Bond et al. [27] studied the use of supercritical CO_2_ dissolution capacity to transport surfactants into the reservoir to generate foam. The results showed that adding surfactants can control CO_2_ in foam formation and improve oil recovery while reducing CO_2_ concentration. Through laboratory experiments, Xue et al. [28] demonstrated the excellent thermal and chemical stability of two surfactants by measuring parameters such as water phase stability, static and dynamic adsorption, CO_2_ solubility, interfacial tension, foam size, and foam viscosity. The two surfactants reduced the adsorption of the sandstone reservoir. They also found that high temperature and CO_2_ dilution by methane reduced foam stability. Foad et al. [29] conducted huff-and-puff experiments on the CO_2_-surfactant system at 80 °C, 4000 psi. The experiment used a non-ionic surfactant, N-NP10c, which belongs to the alkylphenol polyoxyethylene ether class. After eight cycles, approximately 75% of crude oil was extracted. N-NP-10c was 0.15% soluble in supercritical CO_2_ without agents but was 1.76% soluble with agents, an increase of 11 times [24,30]. Bi et al. [30] found that adding ethylene glycol can improve the solubility of surfactants with high polarity in ethanol. Nonionic surfactants can exhibit relatively high solubility in supercritical CO_2_ by the action of alcohol agents. Liu et al. [31] found that bis(2-ethylhexyl) sulfosuccinate sodium salt (AOT) can be dissolved in supercritical CO_2_ with less F-pentanol than ethanol and 1-pentanol because F-pentanol has a “CO_2_-philic” fluorinated alkane chain.

#### 2.1.2. CO_2_ + Gas-Soluble Surfactant + Water (+ Agents)

Based on CO_2_ + gas-soluble surfactant, researchers have improved and proposed the CO_2_ + gas-soluble surfactant + water + (agent) system in which the addition of trace water can form nanoscale aggregates in supercritical CO_2_. Some hydrophilic, extremely polar, or macromolecular solutes can be transported within the aggregates’ core. Since Johnston et al. [32] first confirmed the existence of a supercritical CO_2_ microemulsion, extensive research has been conducted on this subject. Zhang Chao [33] found that as the ethanol concentration in a supercritical CO_2_ microemulsion system increased, the foaming volume increased, then decreased, but remained greater than that without ethanol. The rise in foam half-life with increasing ethanol concentration indicated that ethanol addition facilitated the formation of stable CO_2_ foam by AOT. Cui et al. [34] demonstrated that the tendency of the cloud point pressure of a supercritical CO_2_ microemulsion system to vary with temperature differs depending on the water content. The cloud point pressure of the supercritical CO_2_ microemulsion drops with increasing surfactant AOT content before increasing. Notably, the relationship between AOT content and cloud point pressure is consistent at various temperatures. Some researchers found that [35] alkyl chain alcohols can greatly reduce the cloud point pressure, and the smaller the molecular weight of the alcohol, the better the effect. When the temperature increases, the cloud point pressure increases, which then decreases when the alcohol concentration increases.

The mechanism of surfactant dissolution in CO_2_ can be divided into two categories: (1) the surfactant can directly dissolve in the CO_2_ system, and (2) the addition of surfactant and a small amount of water to CO_2_ to form a supercritical CO_2_ microemulsion system containing polar microwaters. This system is thermodynamically stable, isotropic, and optically transparent. Furthermore, the simple use of a CO_2_ + gas-soluble surfactant system can clog the channel and improve oil recovery efficiency by forming foam in situ with formation water. However, due to the low solubility of CO_2_ in gas-soluble surfactants, the foam performance of the surfactant is limited, hence limiting the flooding effect. Adding alcohol agents can effectively enhance the CO_2_ solubility of a gas-soluble surfactant while promoting the formation of a supercritical CO_2_ microemulsion. In the past, increasing the solubility of supercritical CO_2_ has primarily been aimed at extracting hydrophilic, highly polar, or macromolecular solutes, such as metal ions and proteins, for various applications in chemical engineering and materials science. Therefore, the second method is often used to form a supercritical CO_2_ microemulsion to improve supercritical CO_2_ solubility.

**Table 1 molecules-29-05411-t001:** Progress of two types of CO_2_ foam systems in the field or experiment.

Surfactant	Additives	Research Focus	Results	Conditions	References
AMPHOAM, LDMAA	—	Foam stability at high temperatures and salinity, effect of methane dilution	Good thermal and chemical stability. High temperatures and methane dilution reduce foam stability	120 °C, High salinity	[28]
SURFONIC^®^ N-100, SURFONIC^®^ TDA-9	—	Surfactant effect on rock wettability and CO_2_ huff-and-puff performance	Enhanced rock wettability and improved recovery (75%)	80 °C, 2000–5000 LBS/Sq	[29]
N-P series, ABS, A-S-12, N-NP-7c/9c	Ethanol, Glycol	Solubility and extraction performance of surfactants with CO_2_	N-P-12 with additives showed the highest solubility, stable foam at 125 °C	125 °C	[30]
AOT	Ethanol, 1-pentanol	Solubility in supercritical CO_2_ under different additives and pressures	F-pentanol enhanced solubility and lowered cloud point pressure	52.2 °C, 35 MPa	[31]
AOT, SDS, C8PnEm, C12EmPn	—	Comprehensive performance screening of surfactants and effect of pressure and temperature	Optimal concentration at 0.5%; stability varies with temperature and pressure	40 °C, 15–50 MPa	[33]
AOT	Ethanol	Effect of temperature, water content, and AOT concentration on cloud point pressure of CO_2_ microemulsions	Cloud point pressure changes with AOT content and temperature	45 °C, 19 MPa	[34]
Ls-54	Alcohols (various)	Effect of different alcohols on the phase behavior of CO_2_-containing surfactants	Smaller molecular weight alcohols lower cloud point pressure, temp increases pressure	308.2 K and 318.2 K, 13.67–22.87 MPa	[35]

### 2.2. Evaluation of the System

The CO_2_ foam research based on gas-soluble surfactants is not yet mature. Existing research mainly focuses on indoor experimental and theoretical studies, and only a few field tests exist. The Petroleum Engineering Technology Research Institute of Shengli Oilfield conducted the first on-site test on the application of carbon dioxide gas-soluble foaming agent in China to address the occurrence of gas channeling in the high 89-1 block carbon dioxide flooding reservoir of Chunliang Oil Production Plant. The test successfully formed a carbon dioxide foam underground, effectively blocked the underground channeling channel, and improved the spread area of CO_2_ and oil recovery. According to these studies, gas-soluble CO_2_ foam is mainly injected in two ways: (1) through water-alternating-gas injection, which entails dissolving the surfactant in the gas phase and injecting it into the formation with the gas phase slug [36]. (2) The second approach entails dissolving the surfactant in CO_2_ and injecting the mixture into the formation continuously. During the injection process, the liquid phase is not injected. Foam generation results from the interaction between the CO_2_-carried surfactant and water phase in the formation [23].

The laboratory study of CO_2_ foam mainly includes static and dynamic experiments. Static experiments are always used to investigate the foaming ability and foam stability of surfactants. The surfactant’s foaming ability was analyzed based on the observed foam volume and half-life. Figure 4a [37] shows the static experimental system, which includes a CO_2_ gas source, foam solution tank, and PVT reactor. Through the reactor’s viewing window, the foam volume and foam half-life can be observed to evaluate the surfactant’s foaming ability and foam’s stability. Figure 4b [33] depicts the dynamic experimental system, which mainly includes the sand-filling model, injection, data reading, production, and auxiliary systems. The injection system mainly comprises a CO_2_ gas cylinder, dryer, liquefier, booster pump, sampler, and intermediate container.

Notably, the surfactant’s interfacial properties must be measured using laboratory experiments to evaluate its performance. Figure 5 [33] depicts the experimental system, which mainly comprises a high-temperature and high-pressure vessel with a visual window, pressure, and temperature control system, CO_2_ injection system, motor drive system, and computer data acquisition system. The system could measure temperatures up to 200 °C and pressure up to 20 MPa. The interfacial tension (IFT) between CO_2_ and surfactant solution was measured by automatic droplet shape acquisition and analysis using electric heating. In summary, the laboratory foam injection system is generally divided into three parts: CO_2_ gas source, sample preparation instrument, and surfactant solution. According to different uses, the laboratory research system can be divided into static experiments (determination of foaming performance and foam thermal stability) and dynamic experiments (determination of foam plugging performance). As a result, depending on the experimental purpose, a suitable experimental system and methodologies must be chosen.

In terms of experimental methods, future research should aim to improve both static and dynamic testing protocols for CO_2_-soluble surfactants. Current laboratory methods for testing surfactant foam stability and injectability under high-pressure, high-temperature conditions can be enhanced by incorporating advanced simulation techniques such as molecular dynamics simulations and computational fluid dynamics (CFD) models. These simulations can help predict surfactant behavior before physical experimentation, thereby reducing trial-and-error in the laboratory.

Additionally, high-throughput screening techniques could be developed to accelerate the discovery of new surfactants. This could involve automated systems capable of testing thousands of surfactant formulations under varying conditions of temperature, pressure, salinity, and reservoir heterogeneity, providing rapid feedback on the most promising candidates for further testing. In terms of field testing, mini-pilot tests in controlled environments could offer valuable data on surfactant performance at a fraction of the cost and risk associated with full-scale field trials.

Finally, further improvements in microemulsion and foam generation technologies, such as real-time in situ monitoring of foam behavior during injection, would allow for better control over foam stability and distribution in the reservoir. Utilizing sensor technologies that can track foam formation and movement within the reservoir could provide valuable insights into optimizing foam injection strategies and improving the overall efficiency of CO_2_-EOR processes.

### 2.3. CO_2_ Foam Performance Evaluation Methods

The dissolution, foaming, and displacement of the CO_2_ foam process require a series of evaluation indicators for analysis. To realize a better model for CO_2_ foam performance and effectiveness, the program and equation should be adjusted and improved promptly. Table 2 lists the common indices of gas-soluble CO_2_ foam laboratory research. Using foam volume (*V*) (or height) and foam half-life (*t*_1/2_), the static experimental evaluation method can be used to assess the gas solubility, foaming ability, stability, compatibility, and heat resistance of a surfactant. The macroscopic phase behavior of a surfactant, water, and CO_2_ can also be determined based on the cloud point pressure, after which the stability and solubilization performance of the microemulsion can be determined [33]. Surfactant dissolution in supercritical CO_2_ can alternatively be viewed as the mixing of solvent and solute. The solubility parameter and volume fraction of solvent and solute were used to determine the mixing enthalpy, which was then used to calculate the solubility parameter (*δ*) [38]. The dynamic experimental evaluation method entails analyzing the foam’s plugging ability by measuring the basic pressure difference (∆*P_b_*) and working pressure difference (∆*P_r_*) on both sides of the sand-filled pipe to obtain the resistance coefficient (*R_f_*) and residual resistance factor [33,39,40]. By calculating the residual oil saturation of the core [41], the oil displacement efficiency (*E_D_*) of the core experiment was obtained [39]. The evaluation method for improving sweep efficiency entails using a two-dimensional model or double-tube parallel model for the displacement experiment. Gas injection temperature, gas injection rate, oil production rate, cumulative liquid production, cumulative production degree, and water content should be recorded, after which a map of the relationship between the above data and pore volume or experimental time is drawn and analyzed. In addition, the gas-soluble surfactant can effectively reduce the miscible pressure of CO_2_ because it forms miscible flooding with crude oil. Therefore, the interface performance must be measured using the hanging drop method to examine the change in miscible pressure [42].

Foam rheology requires careful consideration of foam quality and gas volume fraction in the total injected fluid. Steady-state foam can be divided into two state intervals: low mass and high mass [43]. In low-mass regions, shear thinning dominates foam behavior, which can be characterized by an empirical equation, whereas high-mass regions display both shear thinning and shear thickening. The foam behavior in the high-mass area can be analyzed using a model based on the concept of “limit capillary pressure” [43,44,45]. The foam strength increases as the foam mass increases in the low-mass areas. The foam strength peaks at the junction between the high and low foam masses and decreases as the foam mass increases in the high-mass areas. This trend is due to the different behavior of foams in these regions. Assume that the foam size is constant in the low-mass areas; then, as the foam quality increases, the number and apparent viscosity of the foams increase [46]. In addition, a varying range of foam sizes exists [47]. Most current research on foam quality is concentrated in the 40%–95% range because fluidity reduction is infeasible due to foam’s instability outside this range [48]. In addition, advanced models that simulate foam behavior in porous media have also considered the effects of nanoparticle-enhanced stability, which allows for more accurate predictions of foam performance in the field [49,50].

Because it cannot be measured directly, foam viscosity is usually expressed as foam fluidity or apparent viscosity. Foam fluidity is defined as the ratio of effective permeability to apparent viscosity. If the gas phase is continuous, the foam will only reduce the cross-sectional area through which the gas flows, resulting in a drop in relative permeability. If the gas phase is discontinuous, the relative permeability decreases, and the foam also has a high apparent viscosity, which reduces the gas fluidity [51]. Bond et al. [52] first defined the concept of “foam-reducing gas mobility” and expressed it by MRF (mobility reduction factor). Thereafter, the MRF is commonly used to describe foam-reducing gas mobility. MRF can be calculated using the ratio of foam flooding pressure drop to water/gas flooding pressure drop [53]. The apparent viscosity of foam in porous media depends on foam size. The smaller the foam, the more lamellae are transported through the porous medium and the greater the flow resistance [54]. Falls et al. [51] found that if the ratio of foam size to average pore size decreases by two times, the apparent viscosity of the gas increases by an order of magnitude.

In summary, the static and dynamic evaluation methods of CO_2_ foam experiments can effectively evaluate most gas-soluble surfactant foams.

**Table 2 molecules-29-05411-t002:** Evaluation parameters of CO_2_ foam laboratory study.

Evaluation Index	Definition	Equation	Reference
Foam volume (*V*)	The space volume occupied by foam at a certain moment	—	[37]
Foam half-life (*t_1/2_*)	Time taken to reduce the foam volume from the maximum foam volume (*V*_max_) to half the volume at a given temperature	—	[55]
Cloud point pressure	Read the pressure at the critical point when the system becomes turbid	—	[33]
Solubility parameter (*δ*)	Proposed based on the regular solution theory, whose value is the square root of the liquid cohesive energy density, used to characterize the strength of interaction between simple liquid molecules	$\Delta H_{m}=V_{m}\varphi_{1}\varphi_{2}{\left(\delta_{1}-\delta_{2}\right)}^{2}\;$ (1)	[38]
Basic differential pressure (∆*Pb*)	The pressure difference generated at both ends of the core when water and non-condensable gas are injected into the core at the experimental injection rate and gas–liquid ratio	—	[39]
Working pressure difference (∆*Pr*)	Pressure difference at both ends of a foam solution of a certain concentration and a non-condensable gas injected into a core at the same injection rate and gas–liquid ratio as the basic pressure difference measured	—	[39]
Resistance factor (*Rf*)	At a certain temperature, the ratio of resistance pressure difference (∆*P_r_*) to basic pressure difference (∆*P_b_*)	$R_{f}=\frac{∆P_{r}}{∆P_{b}}\;$ (2)	[40]
Residual resistance factor	Ratio of pressure difference between two ends of subsequent water flooding to that before foam injection and after foam injection	—	[33]
Residual oil saturation (*Sor*)	Percentage of residual oil in rock pore volume	$S_{or}=\frac{W_{or}}{V_{p}\rho_{oil}}\;$ (3)	[41]
Displacement efficiency (*E_D_*)	The ratio of produced oil to crude oil in the range of underground displacement	$E_{D}=\frac{\sum_{i=1}^{n}\frac{W_{i}}{\rho_{oil}}}{S_{or}V_{p}}\times 100\%\;$ (4)	[39]
Interfacial tension	Shrinkage capacity at unit length liquid interface	—	[33]
Foam quality (*%*)	Gas volume fraction in foams	$Foam\;Quality(\%)=\frac{gas\;volume}{gas\;volume+liquide\;volume}\times 100\;$ (5)	[56]
Mobility reduction factor (*MRF*)	Ratio of foam flooding pressure drop to water/gas flooding pressure drop	$MRF=\frac{\left(\frac{\Delta p_{foam\;flood}}{Q_{foam\;flood}}\right)}{\left(\frac{\Delta p_{water/gas\;flood}}{Q_{water/gas\;flood}}\right)}\;$ (6)	[57]
Apparent foam viscosity (***μ**_app_*)	Ratio of shear stress to shear rate of foam under certain velocity gradient	$\mu_{app}=\frac{k}{\mu_{t}}\cdot \frac{\Delta p}{L}\;$ (7)	[58]

## 3. The Theory of CO_2_-Soluble Surfactant Foam and the EOR Mechanism

### 3.1. Definition of Foam

Foam is usually defined as a dispersion of gas in a continuous liquid phase [51,59,60,61]. A “lamella” separates the gas phase and makes it discontinuous in a liquid film. The lamellae are thin liquid films with interfaces on both sides of the liquid phase (Figure 6) [62]. In other words, they are thin, free aqueous retention layers surrounded by gas on both sides. The liquid phase usually contains surfactants, which stabilize the lamella via surfactant adsorption at the gas–liquid interface [60]. Surfactant molecules are generally adsorbed on both sides of the liquid film to stabilize the film. The liquid film thickness is only a few microns or even only a few nanometers. When the film is connected to other lamellae, the area can be expanded to a few square meters [63]. The liquid phase may also contain macromolecules or solid particles as alternatives to surfactants. As shown in Figure 7 [61], the bottom of the bulk foam structure is the liquid phase, and the top is the gas phase. The gas phase separates from the lamellar liquid phase through a two-dimensional interface at the microscopic level.

### 3.2. Surfactant Solubility in CO_2_

Because the solubility of gas-soluble surfactant in CO_2_ plays a decisive role in its displacement effect, several studies have been conducted in this regard. Rossen et al. [64] qualitatively determined the solubility of more than 130 commonly used surfactants in supercritical CO_2_, including ionic and nonionic surfactants. The results showed that these surfactants were basically insoluble or slightly soluble in supercritical CO_2_. Consan et al. [65] also found that most surfactants are insoluble or slightly soluble in supercritical CO_2_. These common surfactants failed to dissolve in CO_2_ because the interaction between the hydrophobic chains of surfactants was stronger than that between the CO_2_ and hydrophobic chains. To address this issue, Hoefling et al. [66] improved the surfactant by introducing functional groups with low polarity, low solubility parameters, and Lewis bases into the CO_2_-philic tail chain. Gas-soluble surfactants should meet the following three conditions [67,68,69,70]: (1) The hydrophobic tail chain with the CO_2_-philic functional group must be able to improve surfactant compatibility with CO_2_ and must have a lower cohesive energy density to reduce the interaction among surfactants. (2) Surfactant hydrophilic-linked double-tail chains or branched chains on the hydrophobic tail chain should be able to increase steric hindrance. (3) Greater flexibility of the surfactant molecular chain should translate into higher surfactant solubility in CO_2_. The glass transition temperature primarily determines the flexibility of the molecular chains. The flexibility of a molecular chain increases as its glass transition temperature decreases. External factors also influence solubility. In general, while solubility increases with increasing pressure, it decreases as temperature rises.

Surfactants can dissolve directly in CO_2_ when a small amount of water is added to form a supercritical CO_2_ microemulsion system. Supercritical CO_2_ microemulsion is a water-in-CO_2_ (W/C) emulsion. The continuous phase is supercritical CO_2_, and water is the discontinuous phase. This microemulsion usually comprises supercritical CO_2_, surfactant, and water. Figure 8 [34] shows the structure diagram of the supercritical CO_2_ microemulsion and its formation process. The surfactant hydrophilic chain is inward, forming a nanoscale polar “micro pool” that can hold water molecules in the CO_2_ continuous phase. The surfactant hydrophilic CO_2_ tail chain extends in the CO_2_ continuous phase. The droplet size of the microemulsion is generally ten to several hundred nanometers [71,72,73]. The ratio of solubilized water to the number of surfactant molecules in the microemulsion (denoted as W_0_) reflects the size of the microemulsion water content—an important parameter that reflects the microemulsion characteristics.

Supercritical CO_2_ microemulsions have the following main characteristics [74,75]: (1) Supercritical CO_2_ microemulsion is a thermodynamically stable system. Macroscopically, all directions are uniform, clear, transparent, and have a good dispersion. (2) The aqueous phase of the microemulsion core has two states: free water and bound water. The bound water closely interacts with the surfactant hydrophilic chain in the microemulsion, making the bound water close to the polar surface of the microemulsion core. Free water exists at the center of the microemulsion core, and its water molecules are highly hydrogen-bonded. (3) The continuous phase of CO_2_ has very low viscosity and interfacial tension, and the specific surface area of the water core of the microemulsion is large, indicating that a very large contact area exists between the continuous phase CO_2_ and the dispersed phase. (4) Although the continuous phase of the supercritical CO_2_ microemulsion is nonpolar CO_2_, the aqueous phase of its core is polar. Therefore, supercritical CO_2_ can solubilize fat-soluble organic compounds and water-soluble polar compounds but cannot dissolve high-ion compounds [76]. The ability of surfactants to form microemulsions in supercritical CO_2_ is closely related to their molecular structure [77]. Equation (8) is the ratio of the cohesive energy of the surfactant functional groups at the interface of the microemulsion aggregates. It can be used to predict whether a surfactant can form a microemulsion in supercritical CO_2_ or water.
(8)$$R=\frac{A_{co}-A_{oo}-A_{ll}}{A_{cw}-A_{ww}-A_{hh}},\;$$
where *A_co_* represents the interaction between the CO_2_-philic group in the surfactant and CO_2_. *A_oo_* represents the CO_2_–CO_2_ interaction. *A_ll_* represents the interaction between the CO_2_-philic groups in the surfactant. *A_cw_* represents the interaction between the hydrophilic groups and water in the surfactant. *A_ww_* represents the interaction between water molecules. *A_hh_* represents the interaction between the hydrophilic groups in the surfactant.

This equation essentially compares the cohesive energy of surfactant molecules in the CO_2_ phase with that in the water phase. A higher value of RRR indicates that the surfactant is more likely to dissolve in supercritical CO_2_, while a lower value suggests that the surfactant will more readily dissolve in water or other polar solvents.

The practical use of Equation (8) lies in its ability to provide a theoretical basis for predicting the solubility of different surfactant formulations in supercritical CO_2_. By calculating the cohesive energy parameters based on molecular dynamics simulations or experimental data, researchers can estimate the likelihood of a surfactant dissolving in CO_2_ without the need for extensive trial-and-error experimentation. This can save significant time and resources in the development of new CO_2_-soluble surfactants.

For example, if A_CO_ (the interaction between the CO_2_-philic group and CO_2_) is much larger than A_CW_ (the interaction between the hydrophilic group and water), the surfactant is likely to be more soluble in CO_2_. Conversely, if the interaction energies between the surfactant and water are stronger, the surfactant may show better solubility in aqueous environments rather than in supercritical CO_2_.

While Equation (8) provides a useful framework, its application is limited by the accuracy of the input parameters (interaction energies), which are often derived from empirical or estimated data. Additionally, the equation assumes a simplistic interaction model that may not fully account for complex molecular interactions in heterogeneous reservoir environments. Therefore, future research should focus on refining the calculation of these interaction energies, potentially using more advanced techniques like quantum chemical calculations or molecular dynamics simulations, to improve the predictive accuracy of the model.

When supercritical CO_2_ dissolves some solutes, the dissolution process can actually be regarded as the solvent–solute mixing process. The solubility parameters and volume fractions of solvents and solutes affect the mixing enthalpy during this process [38], and the relationship is shown in Equation (9).
(9)$${\Delta H}_{m}=V_{m}\varphi_{1}\varphi_{2}{\left(\delta_{1}-\delta_{2}\right)}^{2},$$
where Δ*H_m_* represents the mixing enthalpy of the system; *φ*_1_ represents the volume fraction of the solvent; *φ*_2_ denotes the volume fraction of the solute; *v_m_* represents the molar volume of the solution; *δ*_1_ represents the solubility parameter of the solvent; and *δ*_2_ represents the solubility parameter of the solute.

The change in Gibbs free energy in the dissolution process is shown in Equation (10).
(10)$${\Delta G}_{m}=\Delta H_{m}-T\Delta S_{m},\;$$
where Δ*G_m_* represents the change in the Gibbs free energy of the system; Δ*H*_m_ represents the mixing enthalpy of the system; Δ*S_m_* represents the mixing entropy of the system; and T represents the temperature of the system.

In general, the dissolution process tends to disrupt the molecular arrangement in the system, so Δ*S_m_* > 0. The dissolution process must satisfy Δ*G_m_* < 0, that is, Δ*H_m_* − *T*Δ*S_m_* < 0. Therefore, to achieve the condition of spontaneous dissolution, Δ*H_m_
*should be as minimal as possible. δ_1_ and δ_2_ should be as close as possible to Equation (9), which is the principle of similar solubility parameters [78].

The solubility parameter is currently an important measure of substance compatibility. To choose a reasonable and effective solvent, the solubility parameter is often used to evaluate the solvent. The traditional solubility parameters can be measured by experimental [79] or theoretical estimation methods [80]. However, the solubility parameters of a CO_2_ supercritical fluid require a large number of targeted measurements under different conditions because they are susceptible to temperature and pressure. Some researchers have proposed some empirical equations, but these equations can only be qualitatively analyzed in most cases [81]. Molecular dynamics simulation can also be used to investigate solubility parameters. Vimon et al. [82] predicted the solubility parameters of 51 common compounds by molecular dynamics simulation and quantum chemical semi-empirical molecular orbital, combined with a multivariate minimum variance regression method. Xia et al. [83] calculated the solubility parameters of 17 polar, nonpolar, and hydrogen-bonded organic solvents by the molecular dynamics method. The calculated results corroborated the experimental values.

Notably, the addition of alcohols as agents facilitates surfactant dissolution in supercritical CO_2_ and promotes the formation of supercritical CO_2_ microemulsions. Ethanol can adsorb at the oil–water interface. This phenomenon can reduce the interfacial tension, adjust the critical micelle concentration (CMC) and hydrophilic balance value, and increase microemulsion stability [84,85]. Some theories suggest that alcohol molecules can permeate into the surfactant tail chain to reduce the interaction between surfactants [86,87]. Zhang [33] simulated the solubility parameters of a supercritical CO_2_/alcohol agent system by molecular dynamics simulation, providing basic data for characterizing the influence of alcohol agents on supercritical CO_2_ solubility. However, the role of alcohol agents in the surfactant system is still not sufficiently clear. The interaction mechanism between hydrocarbon surfactants/CO_2_/agents needs to be continuously improved. Further research is also needed to develop a more convenient and accurate method for calculating the solubility parameters of supercritical CO_2_.

### 3.3. Foam Generation and Stability

#### 3.3.1. Foam Generation

Foam formation in porous media can be defined as the formation of a new foam lamella. Foam formation is a complex process, and its formation mechanism has basically reached a consensus, which can be divided into three cases: snap-off, lamellar division, and leave-behind [88]. The first two cases produce strong foams, and the latter produces weak foams. These three mechanisms are the basis for understanding foam behavior in porous media, so they are introduced separately.

When Radke et al. [89] first proposed the snap-off mechanism, they believed that snap-off was the most important mechanism for direct visual observation of foam formation. Kovscek et al. [90] developed a fluid dynamics theory to describe the horizontal aggregation of liquid in narrow and angular pores under the action of a pressure gradient. They developed a statistical network model to describe the average rate of foam formation by detachment in porous media. As shown in Figure 9 [90], the foam will snap off when the foam curvature difference between the front part of the foam and the contraction part is enough to cause the liquid flow to shrink back. Foam detachment can be divided into the following three cases: (1) When the upstream pressure exceeds the capillary pressure, the gas phase begins to invade the pore throat. The foam front enters the downstream pore, and the liquid phase rushes into the throat due to the capillary pressure gradient, after which the curvature of the foam front decreases with expansion. The rapidly flowing liquid phase forms a liquid ring until the foam snaps off. (2) When foams block the throat, the ensuing liquid pressure gradient drives the accumulated liquid upstream of the throat to clamp smaller foams. (3) The foam separates when the foam breaks away from a long, straight channel. These three cases may occur in the absence of a surfactant. The static condition for snap-off is that the ratio of the pore throat to the channel is approximately 1:3, depending on the specific shape of the channel cross-section. This ensures that when the interface moves from the pore throat to the channel, the capillary pressure at the front of the interface is less than the capillary pressure at the throat [62]. Some studies have shown that crude oil can affect foam snap-off. The snapped-off oil in the pores downstream of the pore throat will hinder or prevent the snapping-off of foams. This hindrance may be the result of insufficient surfactant concentration in the foam to prevent film rupture or an unstable oil phase surface of the pseudo-emulsified film.

After the foams continue to build to create a large plate, they reach branching points during the migration process, which results in lamellar division because the large plate diffuses in two directions under the action of capillary force [89]. The process of lamella division requires a large foam size, capillary force, and protruding rocks. Figure 10 shows the main mechanism [90]. Chambers et al. [91] also observed similar results in the foam flooding experiment of a micro-etching glass model. In porous media, lamella division occurs if the average size of foams is equivalent to or greater than the average size of pores. This means that lamella division occurs only when the lamella is present, so it is not the primary mechanism for foam formation. However, some studies have shown that lamella division is the main mechanism of foam generation [90]. Furthermore, lamella division will not occur if the captured foam has occupied any of the channels. The new lamella is smaller than the original lamella when lamella division occurs. The frequency of lamella division is a function of the flow foam density, gas velocity, foam size, branching point, and capillary pressure [51].

When the gas flows through an adjacent pore throat, the non-wetting phase replaces the wetting phase, and the two wet surfaces bridge together to form a lamella—the leave-behind occurs. A new lamella is formed at the pore throat between the two protrusions [89]. As shown in Figure 11 [92], unlike the film perpendicular to the flow generated by snap-off and lamella division, the film is parallel to the flow that the leave-behind generates. The leave-behind generates a weak foam that cannot move initially. The effect of leave-behind is much lower than that of the snapping-off mechanism in terms of gas diversion capacity.

#### 3.3.2. Foam Stability

After the formation of new foams, the liquid film of the lamella will be thinner due to the disturbing force. The lamellae are unstable and tend to merge when the thinning process continues until the lamellae reach a critical thickness [93]. While foam stability in porous media determines the oil displacement efficiency of foam, foam coalescence is the main factor affecting foam stability. Therefore, discussing the mechanism of foam coalescence is necessary.

Foams generally coalesce in porous media for the following two reasons: capillary–suction coalescence and gas diffusion [91]. The capillary–suction coalescence is widely considered the main mechanism of film rupture. Liquid saturation, rock permeability, and surfactant concentration affect capillary force. Gas diffusion affects the capture of foams, and it is less common in porous media because the radius of the foam curvature is related to the pore throat and pore volume rather than the foam volume. The experiment shows that as the pore diameter of the porous medium is closer to the actual pore diameter of the reservoir, the foam duration is longer, indicating that “foam oil” will exist for a long time during the development of the reservoir.

The effects of varying film thicknesses on the lamella are also distinct. The thinning of relatively thick films (>100 nm) is primarily influenced by capillary force and gravity [94]. In addition to gravity, the pressure gradient between the center and edge of the lamella primarily influences the thinning of horizontally thick films, as shown in Figure 12 [51].

The thinning rate of the thick film is described in Equation (11) [60]. The formula shows that the film drainage rate is inversely proportional to liquid viscosity. Another factor that affects the stability of thick film foams is the viscoelasticity of the foam surface—the Marangoni effect [62]—which reduces the drainage rate of the films because it can prevent liquid from flowing out from the high surface tension region of the film. The drainage time is proportional to the enhanced Marangoni effect [95].
(11)$$-\frac{dh}{dt}=\frac{2h^{3}\Delta P}{3\eta R^{2}},$$
where −*dh/dt* represents the film thinning rate (m/s); *h* represents the instantaneous thickness of the film (m/s); Δ*P* represents the pressure difference between the center and edge of the sheet (Pa); *η* represents liquid viscosity (Pa·s); and *R* represents the radius of foam (m).

The film drainage rate deviates from the calculation results of Equation (11) when the thin film thickness is less than 100 nm, mainly due to the enhanced separation pressure. Separation pressure retards film thinning through the interaction of long-range electrostatic repulsion (*Π_elec_*), short-range steric hindrance (*Π_steric_*), and short-range dispersion force (*Π_vdW_*). The overlapping electric double layer around the two lamella interfaces produces *Π_electro_* [51]. Its strength depends on the electrolyte concentration in the aqueous phase and the charge density at the gas–liquid interface. The hindrance to the film thinning interaction between the lamella interfaces and film structures produces *Π_steric_*. The structures that provide resistance may be entangled polymers, layered oil droplets, layered spherical micelles [51], or entangled columnar micelles [96]. *Π_vdW_* is usually an instantaneous induced dipole attraction between two particles located at the lamella interface. Its strength depends on the material density of the adjacent phase. However, the separation pressure is mainly aimed at the behavior of static lamellae, and it cannot describe the dynamic behavior. Khatib et al. [45] found that a flowing foam has a limited capillary pressure (*P_c_**), which means that a critical water saturation (*S_w_**) exists. As shown in Figure 13 [63], lower water saturation leads to higher capillary pressure. When *P_c_* exceeds *P_c_**, the foam becomes unstable. The stability of a flowing lamella determines the stability of a flowing foam without oil. When the lamella becomes thin to a critical thickness, it breaks.

It may inhibit foam formation and reduce foam stability when the foam contacts different phases (including the oil phase). The influence of the oil phase on foam stability depends on the structure of the oil, surfactant, and water phases [97], which can be expressed by three interaction parameters: diffusion, entry, and bridging coefficient.

Diffusion coefficient (*S*): If the affinity of the oil to the new phase is strong, the oil diffuses at the gas–liquid interface to form a film when the oil contacts the gas–liquid interface, but if the affinity of the oil to the new phase is weak, the oil will form small droplets at the interface. The diffusion coefficient is a function of the interfacial tension of oil, gas, and surfactant relative to each other, as shown in Equation (12). If the expansion coefficient S is positive, the oil diffuses on the foam. This often leads to foam instability.
(12)S=γS⁄G−γSO−γOG

Entry coefficient (*E*): An insoluble reagent may enter the gas–liquid interface when the insoluble reagent (such as oil) disperses inside the foam film. A new gas–oil interface will be generated if the entry coefficient is positive after the oil has entered the gas–liquid interface and some oil–surfactant and gas–surfactant interfaces are destroyed, resulting in interfacial film instability. Oil dispersion in the film does not necessarily lead to foam instability [61]. Theoretically, if the entry coefficient is negative, the oil will flow out of the film surface and will not render the foam unstable. When the number of films is small, the foam is the most stable [97].
(13)E=γS⁄G+γS⁄O−γOG

Bridging coefficient (B): The oil may bridge adjacent foams if the oil enters the gas–liquid interface but does not diffuse. When the bridging coefficient is positive, the film is unstable, whereas when it is negative, the film is stable. The equation is as follows:(14)$$B={\left(\gamma_{S⁄G}\right)}^{2}+{\left(\gamma_{S⁄O}\right)}^{2}+{\left(\gamma_{O⁄G}\right)}^{2}$$

In addition, factors such as temperature, surfactant concentration, salinity, and solid particles affect foam stability.

In general, foam stability decreases with increasing temperatures. The drainage time decreases, and foam rupture speed increases when temperature increases [95,98,99]. Below the CMC, the increase in concentration results in a decrease in interfacial tension and stabilization of the foam. Nikolov et al. [100] realized improved foam stability when the surfactant concentration markedly exceeded the CMC. Depending on the type, concentration, salinity, and divalent ions of surfactants, salinity may enhance or suppress foam stability. The effect of pH on foam stability is relatively small when the concentration is above the CMC [31].

Solid particles may enhance or decrease foam stability. If the particles are not hydrophilic, their aggregation at the foam interface can enhance the film’s mechanical stability and can also increase the stability by increasing the overall viscosity when dispersed [61]. Nanoparticle-stabilized foams have shown promising results in improving foam stability, especially under harsh reservoir conditions. Studies have demonstrated their ability to maintain foam integrity over extended periods, even in unconventional oil recovery scenarios [101,102,103]. Nanoparticles play a role in foam stability mainly by increasing the mechanical strength of the lamella, increasing the maximum capillary pressure, and forming a network structure [104]. The adsorption of nanoparticles on the lamellae is one of the main mechanisms of foam stabilization by nanoparticles. Compared with surfactants and polymers, nanoparticles have higher separation energy and specific surface energy due to their small size [105]. Therefore, nanoparticle adsorption on the gas–liquid film is irreversible [106] and can prevent gas diffusion and foam coalescence. Nanoparticles with moderate hydrophobicity tend to adsorb on the gas–liquid interface and are more stable in the membrane [107]. Therefore, moderately hydrophobic nanoparticles have better foam stability than extremely hydrophobic or extremely hydrophilic nanoparticles. In addition, the interaction between nanoparticles is another important mechanism affecting foam stability. There may be three structures—single layer, thick multilayer, and network structure—when nanoparticles are dispersed in a foam solution [108,109,110]. While the network structure promotes the stability of maximum capillary pressure and separation energy and enhances the energy demand of foam coalescence [111], it prevents the film from thinning and reduces gas diffusion and liquid discharge, thereby preventing small foams from merging into large foams and prolonging the existence time of the foam system [112,113].

During the channeling process, when carrying surfactants, the CO_2_ phase interacts with the water phase in the pore roar to generate foam. Therefore, the crossflow of CO_2_ favors the acceleration of foam generation in the core with high permeability, making the foam plugging effect faster. The surfactant carried in the CO_2_ phase facilitates foam regeneration after the foam ruptures during foam migration, thereby prolonging the foam stability time [33].

### 3.4. Flow and Rheological Model of Foam

CO_2_ foam with good stability is required in the application process. Recent studies have advanced the understanding of foam stability through visualization techniques, showing real-time foam behavior and enhancing predictions of foam stability [114,115]. Compared with the unsteady foam flow study based on relative permeability and viscosity, the steady foam study prioritizes foam fluidity [53]. The sole function of liquid saturation that influences foam fluidity is foam size and structure [116,117]. These two parameters are affected by many factors, such as foam properties, gas content in foam, pore structure, medium heterogeneity, surfactant system, capillary pressure, flow rate, and non-wetting [51,53].

When the pore size of a porous medium exceeds the diameter of a single foam, the foam is referred to as bulk foam [118], and multiple foams then accumulate in the pore space. As shown in Figure 14a [94], while the bulk foam features easily separable spherical foams when the gas content is low, it features polyhedral foam (also known as dry foam) when the gas content is high. However, the average foam size is usually larger than the pore size; that is, a single foam occupies more than one pore space [63]. Foam in porous media typically comprises continuous and discontinuous foams. Discontinuous foam is divided into flowing and retention foams. As shown in Figure 14b [119], the discontinuous foams occupy large and medium pores. Flowing foam usually occupies the largest pores, whereas retention foam occupies small and medium pores. Foam mobility depends on the gas pressure gradient and foam size. Flowing foam and retention foam are dynamic.

Foam reduces liquidity by reducing effective gas permeability and increasing gas flow resistance. Both flowing and retention foams can cause flow resistance. The lamellae reduce the effective permeability of the gas flowing through the connected channels by the required pressure of the shrink pore throat. The effective yield stress of the lamella to the gas will also capture a considerable part of the gas under a high-pressure gradient, thereby hindering gas flow [51,59,120]. The retention foams block the pores to reduce the channels available for gas flow and lower the effective permeability of gas. The foam structure dominates foam rheology, and the porous media regulates the foam structure through capillary pressure when the foam flows in the porous media. The viscous shear stress of the film between the hole wall and the lamellae increases the apparent viscosity, as shown in Figure 15 [121]. When the foam flows, the wetting film covering the pore wall interacts with the lamella. The conditions of equilibrium between the two films are the primary determinants of foam equilibrium. The thermodynamic properties of the two films may dominate foam behavior. This effect is particularly pronounced in strong foams, where the wetting film and lamella limit all foams. Because viscosity and capillary force affect foam flow at any time, foams in the mobile phase will exhibit an obvious dragging phenomenon when passing through the contraction pore throat, resulting in an irregular lamella movement. When the lamellae reach the contraction portion of the channel, their movement accelerates, and their centers become convex backward. However, when the lamellae separate from the pore throat, their movement slows, and their centers become convex forward. When there is constant pressure, the lamellae do not generally move steadily but move alternately by sliding and stopping. This movement is known as the stick–slip motion [122].

Numerous researchers have examined the simulation of foam behavior. Because the factors influencing foam fluidity are numerous and complicated, it is difficult to precisely forecast foam behavior. However, some researchers have proposed many methods to simulate foam flow in porous media. Most models change gas fluidity when foam is generated. These foam models can be roughly divided into four categories [123], empirical or semi-empirical models (empirical changes in gas mobility), theoretical models of separated-phase flow (including critical capillary force model [124], pore network model [125], foam overlay and drainage model [126], and foam structure evolution model [127]), seepage (statistical network) model [128], and mechanical model (population balance model) [129]. The following introduction focuses on empirical or semi-empirical modeling and mechanical modeling methods.

#### 3.4.1. Empirical or Semi-Empirical Models

Because the empirical or semi-empirical model is not a function of the foam structure and gas fluidity but a function of the steady-state gas flow, it is necessary to empirically correct the relative permeability and viscosity of the gas separately or simultaneously. To characterize the function of liquidity reduction, researchers have incorporated the micro-foam process into empirical or semi-empirical models. As a result, no additional conditions are required for the continuity equation or the rate equation of all phases. Two popular empirical models are as follows:(1)STARS^TM^ foam model

STARS^TM^ is a commercial foam model developed by the Computer Modeling Group in Calgary, Canada [130]. The basic assumption is that foam formation and coalescence occur rapidly compared with the flow, so the foam exists whenever the gas and surfactant aqueous solution coexist. The concept of the liquidity reduction factor is proposed as a weighting factor to determine the relative permeability of gas to different foam strengths. The equation is as follows:(15)$$k_{rg}^{f}=k_{rg}^{o}S_{w}*FM$$

The dimensionless interpolation factor FM is in the range of 0–1, where 0 represents a very strong foam and 1 represents no foam. The influencing factors of the value are as follows:(16)$$FM=\frac{1}{1+fmmob*F1*F2*F3*F4*F5*F6*F7}$$

In the formula, *FM* represents the liquidity reduction factor varying between *FM* = 1 (no foam) and *FM* < 1 (strong foam). *fmmob* represents the reference foam fluidity reduction factor. *f_1_* represents the effect of surfactant concentration. *f_2_* represents oil resistance. *f_3_* represents the influence of the capillary number. *f_4_* represents the effect of capillary number on foam formation. *f_5_* represents the oil component impact. *f_6_* represents the salinity effect. f_7_ represents the dry-out effect.

(2)UTCOMP foam model

The UTCOMP model is a foam miscible flooding model. Liu [131] and Chang et al. [132] proposed the UTCOMP model modified based on the “fixed-P_c_*” model. The “fixed-P_c_*” model is a stable state in which layers are maintained in a local equilibrium between foam formation and coalescence when the capillary force dominates the foam structure and gas flow. It describes the local equilibrium state of strong foam. The basic assumption is that the foam structure only adjusts the fluidity of the foam itself and the gas according to demand. Thus, the water saturation stabilizes at *S_w_**, ensuring that the critical capillary pressure (*P_c_**) is independent of the gas–liquid flow rate [133,134]. Therefore, the model does not describe the slow formation or coalescence of foams, nor does it describe a flow rate-dependent process. In general, *P_c_** is affected by factors such as rock properties, surfactant formulation, concentration of each component, and temperature. In some cases, flow rate and foam quality also affect *P_c_** [45]. The surfactant concentration and oil saturation can form a smooth function curve that affects foam strength based on the continuous behavior of foam at the surfactant concentration of *C_s_** [135]. *S_w_** increases with increasing flow rate, and *P_c_** increases with decreasing permeability [45,133]. UTCOMP only represents the effect of foam by changing the relative permeability (*k_rg_*) of the gas for easy application.

In the early UTCOMP foam model [136], there are two additional conditions, *S_g_* > *S_g_^lim^* and *S_o_* < *S_o_^lim^*. In the current UTCOMP model, if the surfactant exists, its concentration surpasses a specific value (*C_S_**), and the water saturation also exceeds a specific threshold (*S_w_**), the foam will then be formed. Make the following changes to $k_{rg}^{f}$ in the model:

If *S_w_* ≤ *S_w_** − *ε* or *C_S_* < *C_S_**, then
(17)$$k_{rg}^{f}=k_{rg}$$

If *S_w_** − *ε* ≤ *S_w_* ≤ *S_w_**+*ε* and *C_S_* ≥ *C_S_**, then
(18)$$k_{rg}^{f}=\frac{k_{rg}}{1+\frac{\left(R-1\right)\left(S_{w}-S_{w}^{*}+\varepsilon \right)}{2\varepsilon }}$$

If *S_w_* > *S_w_**+*ε* and *C_S_* ≥ *C_S_**,then
(19)$$k_{rg}^{f}=\frac{k_{rg}}{R}\;$$

The foam exhibits shear-thinning behavior in the low-mass region after the gas flow rate is modified by the foam parameter (*R*).
(20)$$R=R_{ref}{\left(\frac{u_{g}}{u_{g,ref}}\right)}^{\delta -1}$$

Shi [137] introduced two other new parameters into the model to study the change trend in foam fluidity in low-quality areas on this basis. Jose [138] comprehensively analyzed the research situation of high-quality area and low-quality area [139] models and then proposed a unified “*f_g_** model” for the two areas.

The foam model in the fluid-flowing models of STARS^TM^ and UTCOMP can also be called a heuristic model or local population balance model because they are the methods between experience and population balance. This model keeps the simplification of the former and avoids the complex calculation of the latter. However, due to the use of case-specific methods, empirical models generally lack versatility.

#### 3.4.2. Population Balance Model

Calculating mass and energy transfer in porous media enables researchers to add foam to reservoir simulation using the population balance method—a mechanical method developed based on the concept of tracking the number of foams. This method can quantify the relationship between foam fluidity and structure while separating and defining the formation and rupture mechanism of liquid film or lamella of foams [51,59,140]. The population balance model, which can be subdivided into the population balance model and the local population balance model, is the theoretically most comprehensive foam model framework. The population balance model is a method of tracking the change in the foam structure (*n_f_*) based on mass conservation [129]. The *n_f_* is sometimes referred to as the lamella density of the flowing foam in order to distinguish it from the foam structure (*n_t_*) of the trapped foam [92].

However, a significant number of parameters of laboratory research and field data are difficult to precisely integrate into the model, limiting the population balance method’s use in practice. The cost of directly solving the population balance equation is too high, and a divergence problem exists. Therefore, some scholars have proposed a method to obtain foam density by the number of foams or local balance. This method assumes that foam formation and coalescence are relatively fast compared to the time scale of foams passing through the porous media, and the dynamic equilibrium of the local foam structure is achieved by equalizing the local foam formation (*R_g_*) and coalescence (*R_c_*) rates [44,141,142]. The model is multiphase and multicomponent, which can represent mass transfer between phases and foam dynamics. In addition, the model can predict a wider behavior of the foam dispersion process under different experimental conditions.

The general form of the population balance model is Equation (21) [143]. From left to right is the cumulative term of flowing and trapped foams, the flow rate of flowing foams, the source term of the net generation term and the foam. *S_g_* and *S_gt_* are related to *S_gf_* (Equation (22)). In the equation, *X_f_* is the flow gas fraction, and *X_t_* is the retention gas fraction. The relationship between *X_t_* and the maximum retention gas fraction *X_t,max_* is shown in Equation (23); β is the retention parameter.
(21)$$\frac{\partial }{\partial t}\left(\phi \left(S_{gf}n_{f}+S_{gt}n_{t}\right)\right)+\nabla \cdot \left(u_{g}n_{f}\right)=\phi S_{g}\left(R_{g}-R_{c}\right)+Q_{b}$$
(22)$$S_{gf}=X_{f}S_{g}=\left(1-X_{t}\right)S_{g},\;S_{gt}=X_{t}S_{g}$$
(23)$$X_{t}=X_{t,max}\left(\frac{\beta n_{t}}{1+\beta n_{t}}\right)$$

Bertin et al. [142,144,145] proposed a relation that expresses the change in flow foam density with the physical properties of porous media (porosity, permeability, and capillary pressure), surfactant solution, and flow conditions based on many assumptions. The relation expression is feasible at both pore and core scales. The grain diameter can estimate the amount of porosity per unit volume and can be evaluated simply by the Kozeny–Carman relation. Li et al. [146] established foam models to simulate one-dimensional and three-dimensional foam flow. They did not directly correct the gas relative permeability but proposed an alternative method that considers the interception of gas to simply increase the gas residual saturation when calculating the gas relative permeability. The model calculates the gas’s apparent viscosity through the formula of Friedmann et al. [59], which considers the shear-thinning effect and uses the formula of Bertin et al. [145] to express the foam density (foam structure). Five regions of gas saturation increase are defined and discussed based on this, and the expressions of gas apparent viscosity differ. Chen et al. [147] extended the computing power of previous models to low-quality areas by considering the dependence of foam formation on the existing foams when calculating the foam formation rate because of the limitations of the existing population balance model. They also proposed and implemented a new simplified foam model for the efficient simulation of foam displacement in porous media using the simplified expression of foam generation and the local equilibrium assumption of foam generation and coalescence.

The population balance model obtains the foam structure by solving the partial differential equation based on the general form. The local population balance model obtains the foam structure by solving the algebraic equation of variables obtained by Darcy’s law and the mass conservation equation based on the semi-empirical or population balance model. All the above quantitative equilibrium models solve only two problems: how to obtain *n_f_* (the function of gas saturation, trapped gas fraction, and capillary pressure) and how to change foam fluidity. Due to the different methods of describing slice generation, the expressions of these models also differ. In other words, both the population balance and local population balance models are set to correct only the relative permeability (*k_rg_*) or apparent viscosity (*μ_g_*) in porous media. Most quantitative equilibrium models change the gas’s relative permeability by multiplying with the flow gas saturation and change the gas’s apparent viscosity by $\eta_{app}$ in the capillary. Applying a suitable formula to describe the phenomenon of trapped gas is key to adjusting the relative permeability of gas (*k_rg_*).

Thus far, no single foam model can be applied to all foam experiments in porous media under different conditions using foam mobility modeling technologies. Therefore, fitting laboratory or field data to modeling parameters is particularly important for establishing and validating foam models. Due to the variances across the models, their fitting methods differ. Therefore, establishing appropriate assumptions based on actual data during foam flooding is necessary, as is considering the impact of crude oil on foam stability, liquidity, and other major control factors in the foam flow mechanism.

### 3.5. CO_2_ Foam EOR Mechanism

Foam formation and coalescence affect gas flow during CO_2_ foam flooding. A rheological model can describe the flow characteristics of foam in porous media and can be used to analyze the effect of foam on gas fluidity. However, the CO_2_ foam displacement effect is multifaceted, and its specific mechanism in the experimental process needs to be discussed. CO_2_ foam can effectively reduce CO_2_ fluidity and stabilize the displacement front while significantly improving the crossflow and gravity separation caused by pure CO_2_ injection. The decrease in gas flow is due to the increase in apparent viscosity and trapped gas fraction [53]. Zhang et al. [84] found that AOT dissolution in supercritical CO_2_ can increase supercritical CO_2_ viscosity by three times, and the formation of supercritical CO_2_ foam can increase viscosity by 50–200 times. Foam can produce effective yield stress on gas [115]. Even under a high-pressure gradient, CO_2_ foam can capture a considerable part of gas and hinder gas flow. The static or trapped foam can lower the effective permeability of gas by reducing some porous media available for gas flow, thereby blocking the gas flow channel [59,120]. No maximal pressure gradient may mobilize all foam lamellae during core displacement. This means that the porous media always bound some foams during foam flow [123]. The proportion of foam-bound gas in sandstone is about 85%–99% under steady-state conditions. Foam mobility and gas–liquid interface rearrangement affect gas flow. The additional pressure drop driving the foam at a constant speed exceeds the same volume of liquid, indicating that foam can increase the effective viscosity of the gas phase [148]. The surfactant movement at the gas–liquid interface produces a surface tension gradient, which slows down the foam movement and increases the effective viscosity of the foam [61].

Foam can selectively block heterogeneous media and improve CO_2_ sweep efficiency. As shown in Figure 16 [149], the foam reduces the relative permeability of the gas by blocking the pores of the gas flow and transferring the flow from the area with higher permeability to the untouched area with lower permeability. Foam or gas plugging occurs at the boundary under sufficient permeability differences [150]. The influence of limited capillary pressure on foam lamella is often used to explain the relationship between pressure gradient and gas–liquid flow rate in a high-quality area [45]. The limited capillary pressure is also affected by permeability [141]. Foam diversion is sensitive to permeability in high-quality areas and insensitive to permeability in low-quality areas, but the effect of channeling in low-quality areas improves significantly. This permeability dependence makes foam particularly suitable for improving interlayer heterogeneity [151]. Notably, the competition between gravity (and density difference) and transverse pressure gradient [152] leads to gravity superposition. As the displacement front advances away from the injection well, the pressure gradient and velocity decrease, increasing gravity overpressure. This effect is the same in the cylindrical flow with a fixed injection velocity [137]. Foam can effectively overcome the influence of gravity and improve the CO_2_ sweep efficiency.

Surfactant addition can reduce capillary force and alter rock surface wettability by reducing the oil–water surface IFT. It is beneficial to separate oil from porous media. The oil can form emulsions under reservoir conditions and is easy to displace. As shown in Figure 17 [153], the oil–rock adhesion far exceeds the capillary force that keeps oil on the rock surface in the oil-wet reservoirs. The ability of surfactants to change wettability (rather than reduce IFT) and the diffusion behavior of oil in three fluid (oil, brine, and gas) systems dominate oil displacement [154,155].

In porous media, retention foam can persist for a long time. There is long-term contact of the three CO_2_–film–oil phases when it is in contact with undisplaced oil. The interfacial mass transfer between CO_2_ and oil is strengthened. This makes the mass transfer of CO_2_ to the oil increase and enhances the viscosity reduction and expansion of oil [148,156]. Because high pressure can significantly enhance the mass transfer of CO_2_ into water and oil (n-decane) when a surfactant exists [157], measuring the effect of surfactants on the degree of mass transfer at higher pressures is necessary [158]. Oil will also migrate to the lamella structure of the foam via emulsification dialysis, pseudo-emulsification film thinning, entry, and diffusion. The oil will be emulsified into the aqueous phase, resulting in oil and gas separation of aqueous film as long as the foam diffuses in the crude oil—the pseudo-emulsified film. Notably, if the oil diffuses on the foam, it will diffuse on or penetrate the lamella surface and destroy the lamella inner surface [61]. More crude oil will destroy the foam [159], and CO_2_ will make direct contact with the crude oil.

## 4. Classification and Application Progress of CO_2_-Soluble Surfactants

In this section, we classify the common surfactants in the petroleum industry and describe the screening criteria and parameters of CO_2_-soluble surfactants, after which we introduce the application progress of each type of surfactant (ionic and nonionic surfactants and mixed surfactants) in CO_2_ foam flooding.

### 4.1. Surfactant Classification

Surfactants typically comprise hydrophilic head and hydrophobic tail groups. According to the polarity of the head group, they are divided into nonionic and ionic surfactants. The ionic surfactants can be subdivided into cationic, anionic, and zwitterionic surfactants [160].

Nonionic surfactants have no charge on the head group. Nonionic surfactants typically have high solubility in organic solvents, making them suitable for use in various solvent systems and exhibiting good stability. The common CO_2_-soluble nonionic surfactants are mostly ethoxylates and amine derivatives. Ethoxylates can be divided into branched alkylphenol ethoxylates, branched alkyl ethoxylates (branched ethoxylate alcohols), nonbranched alkyl ethoxylates (linear ethoxylate alcohols), and fatty acid-based surfactants. Among them, the common branched alkylphenol ethoxylates include 2-(2-[4-(1,1,3,3-tetramethylbutyl)phenoxy]ethoxy)ethanol, nonylphenol branched ethoxylated, and tristyrylphenol ethoxylated. Common branched alkyl ethoxylates include polyethylene glycol trimethylnonyl ether, ethoxylated isodecyl alcohol, C_12_–C_14_ fatty alcohols ethoxylated, and GENAPOL(R) X-080. The common nonbranched alkyl ethoxylates include alkyl-(C_10_–C_14_) alcohol, ethoxylated. Common fatty acid-based surfactants include polyethylene glycol monolaurate and polyoxyethylene 20 sorbitan monooleate [24,161]. Amine derivatives mainly include *N*,*N*′,*N*′-polyoxyethylene(10)–*N*–tallow-1,3-diaminopropane (DTM), *N*-oleyl propylene diamine polyoxyethylene ether, coco alkyldimethylamines, and ethoxylated cocoamines, among which ethoxylated cocoamines also belong to nonbranched alkyl ethoxylates [162,163]. In addition, there are some other nonionic surfactants, including propoxylated and ethoxylated dodecanol, oxirane, methyl-, polymer with oxirane, mono(2-ethylhexyl) ether, and modified alkylphenol polyoxyethylene ether. Among them, when DTM and ethoxylated cocoamines are dissolved in an acidic aqueous solution (the pH of an aqueous solution containing CO_2_ is generally 4–5), due to the protonation of the amine head group, the surfactant will change from nonionic to cationic, which has great application potential. The above nonionic surfactants have good solubility under high temperatures, high pressure, and high salt, and their adsorption capacity in carbonate reservoirs is very low [98]. The specific information is shown in Table 3, where N–X is a representative alkyl tail, and the actual surfactant comprises mixed isomers, which is a qualitative characterization in the table.

The ion surfactant has good foaming performance and foam stability. However, due to structural characteristics, the solubility of most ion surfactants in CO_2_ is extremely limited, which limits their application. At present, there are few soluble ionic surfactants for CO_2_, and the common anionic surfactants are AOT and sodium dodecyl sulfate (SDS) [30,84]. Research on cationic surfactants mainly focuses on convertible ethoxylated amine surfactants, such as *N*,*N*′,*N*′-trimethyl-*N*-(tallowalkyl)-1,3-propanediamine (DTM) and ethoxylated cocoamines [164]. Common zwitterionic surfactants are mainly betaines, such as cocoyl amide propyldimethyl glycine (CAB-35) [30] and lauroylamide propylbetaine [165], and N-dodecyl-N,N-dimethyl-3-ammonio-1-propanesulfonate [28].

In addition to the surfactants listed above, there are several more. In the early stages of CO_2_ gas-soluble surfactant research, fluorine- and silicone-containing surfactants were proposed. They include perfluoropolyether ammonium carbonate, double-tailed fluorocarbon sulfate, double-tailed fluorocarbon-hydrocarbon mixed sulfate, fluorine-containing dialkyl phosphate, fluorine-containing AOT homolog, oligosiloxane, and functional silicone [33]. Because these surfactants have small cohesive energies, good molecular chain flexibility, low glass transition temperatures, and small intermolecular interactions [166], they can readily dissolve in CO_2_ [167,168,169]. Fluorine- and silicon-containing surfactants readily solubilize in CO_2_ and exhibit excellent foaming properties and foam stability. Although they are very good CO_2_-soluble surfactants, cost inefficiency and environmental toxicity [169] greatly limit their application.

In addition to using a single surfactant, researchers have mixed cationic, anionic, and nonionic surfactants separately. Because the synergistic effect of different types of surfactants can improve gas fluidity, various surfactant blending systems have been developed, including different ionic surfactant blending systems such as anionic sodium dihexyl sulfosuccinate + cationic benzethonium chloride [170], anionic F-OPT + cationic F-CAT [169], bis(1H,1H,2H,2H-heptadecafluorodecyl)-2-sulfosuccinate and fluorocarbo–hydrocarbon hybrid anionic surfactants [171], as well as nonionic surfactant blending systems such as C_12_NEO_2_ + C_13_EO_12_ [172]. Table 4 lists their specific descriptions.

### 4.2. Application Progress

#### 4.2.1. Nonionic Surfactants

Due to their relatively good solubility in CO_2_, nonionic surfactants have become a research hotspot for CO_2_ gas-soluble surfactants. Xing et al. [161] studied some nonionic surfactants, including alkylphenol ethoxylates, alkyl ethoxylates, and fatty acid-based surfactants. They found that most of these surfactants were soluble in CO_2_ in the range of 0.02–0.06% at 1500 psia and 25 °C, with a certain ability to stabilize foam. The foam produced by branched alkylphenol ethoxylates is the most stable. The emulsions generated by surfactants with longer ethoxylated chains had more droplets and a wider size distribution than their analogs with shorter ethoxylated chains. Bi et al. [30] found that the foaming volume and half-life of CO_2_ foams increased with increasing the number of EO groups (polyoxyethylene groups) in a single molecule of the kind of alkylphenol ethoxylate surfactants. They also found that these surfactants can stabilize CO_2_ in 5 wt% NaCl brine emulsion [24]. Therefore, when the continuously injected CO_2_/surfactant solution is mixed with the formation brine, emulsion or foam may be formed in the reservoir, thus avoiding or reducing the necessity of alternating injection of surfactant aqueous solution during CO_2_ injection. Notably, the π–π stacking of the benzene ring can enhance surfactant stacking at the CO_2_–brine interface and stabilize the water film, resulting in a more stable emulsion generated by alkylphenol ethoxylates than by branched alkyl ethoxylates.

Surfactants, such as alkylphenol ethoxylates or alkyl ethoxylates, have been extensively examined as effective C/W foaming agents at medium and low temperatures (6, 9, and 10 °C). However, at high temperatures, they readily precipitate, and the hydrogen bonds between EO groups and water are weakened and broken, resulting in solvent loss [174,175,176]. Bi et al. [30] revealed that alkylphenol ethoxylate sulfonates obtained by the sulfonation of alkylphenol ethoxylate ethers produced CO_2_ foam with certain stability at 125 °C, with a half-life of 6.54 min. Surfactant hydrophilicity was enhanced after sulfonation, resulting in excellent foam stability. This approach also produced stable CO_2_ foam at higher temperatures, and the foaming ability, foam stability, and temperature resistance of these surfactants were significantly improved. McLendon et al. [177] found that the alkyl chain of the DOW Tergitol NP Series remained unchanged as the polyethylene glycol increased from nine ethylene oxide groups to 12 ethylene oxide groups. The CO_2_-philic polyethylene glycol chains increased, but the CO_2_-phobic hydroxyl groups remained unchanged, resulting in more CO_2_-philic and hydrophilic molecules. In addition, foam cloud point pressure decreased, while the surfactant became more soluble in CO_2_. At a very dilute concentration with approximately 0.10% solubility, branched or linear hydrocarbon tails seemed to have a negligible effect on the CO_2_ solubility of surfactants. Foad et al. [29] studied wettability and found that SURFONIC ^®^ N-100 (0.1 wt%) significantly enhanced CO_2_‘s ability to change the sample wettability from medium to wet, with a water–shale–air contact angle reduced from 118° to 36°.

#### 4.2.2. Ionic Surfactants

Although ionic surfactants may have potentially high cloud point pressures compared to nonionic surfactants, most ionic surfactants have very limited solubility in CO_2_ [178]. This is because supercritical CO_2_ is a non-polar solvent, while ionic surfactants have polar head groups, which leads to poor solubility in non-polar solvents. Some ionic surfactants with CO_2_ solubility developed after years of research.

Sodium alkyl sulfate anionic surfactant has a certain solubility in CO_2_. It can produce CO_2_ foam with a certain stability at 125 °C, and the half-life is about 7.39 min [30]. Under normal conditions, there is no direct dissolution between carbon dioxide and sodium alkyl sulfate. Under supercritical conditions, sodium alkyl sulfate can dissolve in supercritical carbon dioxide. Although anionic surfactants can improve foam stability at lower water saturation compared with nonionic surfactants, thereby reducing residual water saturation and increasing the pore volume of CO_2_ storage [179], their applications also have limitations. They are highly adsorbed on the carbonate surface [180], and some precipitate in hard water or high salt brine [181,182]. Xue et al. [28] studied a sulfonate head-based anion surfactant and found that the surfactant could dissolve in CO_2_ and form stable viscous C/W foam at high salinity (14.6% TDS) and high temperature (120 °C). Viscous foam can still form after CO_2_ is diluted by 20% methane. The presence of high salt concentrations, especially divalent salts, will inevitably lead to the screening of electrostatic interactions between the surfactant head group and mineral surface [183]. Divalent cations can bridge the surfactant head group and mineral surface [184]. Anionic surfactants are usually used in sandstone reservoirs to minimize adsorption. However, various sulfate-containing anionic surfactants are prone to hydrolysis under high-temperature acidic conditions with pH < 4 in sandstone reservoirs in the presence of CO_2_ [185].

Compared with anionic surfactants, cationic surfactants can significantly reduce the adsorption of surfactant components on the surface of carbonates [56]. Cationic surfactants, such as ethoxyamine surfactants, have good solubility and stability in high temperatures and high salt brine, but they are often adsorbed on negatively charged sandstone reservoirs due to electrostatic attraction [186]. The study found that ethoxylated cocoamines can dissolve in CO_2_ and form CO_2_ foam under high temperature (120 °C), high pressure (3400 psi), and high salinity (22% TDS), which effectively reduces fluidity. However, the minimum pressure gradient requires a higher injection rate. A lower injection rate is then required to produce stronger foam due to the shear-thinning effect. Notably, the adsorption capacity of carbonate formation (0.10–0.13 mg/m^2^) is low [186,187]. Although cationic surfactants have good solubility and foaming properties at high temperatures, quaternary ammonium salts are prone to thermal degradation at high temperatures [188,189]. Chen et al. [190] optimized the tail length of trimethyl ammonium cationic surfactant and found that the optimized surfactant can be effectively dissolved in brine with 22% TDS at 120 °C. It can achieve high surfactant adsorption at the CO_2_–water interface (area/surfactant molecule 154 Å^2^), and the interfacial tension is reduced from about 40 mN/m to about 6 mN/m. The above properties make the apparent viscosity of C/W foam at 120 °C to be 14 mPa·s, which is stable in both the crushed calcium carbonate packed bed (75 μm^2^) and the capillary downstream of the bed. In addition, the partition coefficient of surfactant between oil and 22% TDS (255 kg/m^3^) brine is less than 0.15, which will help minimize the loss of surfactant in the oil phase in applications such as EOR and hydraulic fracturing.

Zwitterionic surfactant is a surfactant formed by connecting one or two head groups of surfactants near the hydrophilic head group or hydrophilic head group through a chemical bond. Cocoyl amide propyldimethyl glycine can produce CO_2_ foam with certain stability at a high temperature (125 °C), and the half-life is about 24.96 min. Its foaming performance is good, but the adsorption capacity in the formation is higher [30]. AlYousef et al. [165] found that lauroylamide propylbetaine effectively stabilized the foam at 100 °C, 2000 psi, and >57,000 ppm salt water salinity. The apparent viscosity of the foam was high. With decreased foam quality, the foam viscosity increased. The adsorption experiment results showed that the injected surfactant solution had a recovery rate of 86.56%. Notably, the amount of surfactant adsorbed by rock is about 0.257 mg/g rock, and the adsorption amount of rock is small. Zhou et al. [191] found that the adsorption amount of cocoyl amide propyldimethyl glycine in carbonate was lower (1 mg/g rock) than that in sandstone. Xue et al. [28] studied the N-dodecyl-N, N-dimethyl-3-ammonio-1-propanesulfonate and found that the surfactant dissolved in CO_2_ and stabilized the viscous C/W foam at high temperature (120 °C) and high salinity (14.6% TDS) salt water, even when CO_2_ is diluted by 20% methane.

#### 4.2.3. Fluorine- and Nitrogen-Containing Surfactants

Due to their superior performance, fluorine- and nitrogen-containing surfactants have been the subject of numerous studies in the earliest research on CO_2_-soluble surfactants. Marcio et al. [192] experimentally demonstrated the interaction between fluorine atoms and CO_2_ molecules. According to the Lewis acid–base theory, they considered fluorine to be a Lewis base, and carbon in CO_2_ to be a Lewis acid. By preparing W/C microemulsion with fluorinated AOT homolog surfactants, Eastoe et al. [193] found that the structure of fluorinated surfactants had a significant effect on the microemulsion’s stability. They believed that the fluorine atom could shield fluorocarbon chains well because it has a higher radius than the hydrogen atom. Moreover, the low cohesive energy density, solubility parameter, and polarization ability of fluorinated chains make fluorocarbon chains more compatible with CO_2_. Therefore, fluorinated surfactants can act well on the water–CO_2_ interface and have high surface activity. Eastoe et al. [194] studied the W/C stability of various fluorine-containing AOT homologs to better elucidate the relationship between the structure and properties of these homologs. These surfactants have two tail chains with different fluorination degrees, and the perfluorinated tail chain plays an important role in stabilizing the W/C microemulsion. The main reasons for the good solubility of fluorinated surfactants in CO_2_ are summarized as follows [166]: (1) The cohesive energy density of fluorine-containing surfactants is small. The presence of fluorine atoms can reduce the interaction force between surfactants. (2) Fluorine atoms with high electronegativity can interact with carbon atoms in CO_2_ to increase the interaction force between surfactants and CO_2_. The flexibility of the fluorinated chain is better, and the glass transition temperature is lower. (3) The presence of fluorine affects the acidity of adjacent protons, resulting in specific interactions between these protons and the oxygen atoms of CO_2_. Furthermore, silicon-containing surfactants have high surface activity similar to fluorine-containing surfactants. Silicon-containing molecular chains have good flexibility, low glass transition temperature, and cohesive energy density, in addition to small intramolecular interactions that facilitate the dissolution of siloxane polymers in CO_2_ [195]. Beckman [166] synthesized two functional silicones based on previous studies. The difference in the functional silicones is the carbonyl group on the side chain. They found that adding a carbonyl group to silicon-containing alkyl side chains can greatly improve the solubility of functional silicones. Carbonyl group addition reduces the intermolecular force of functional silicone and improves surfactant solubility. In this case, CO_2_, as the Lewis acid, reacts closely with the carbonyl group as the Lewis base. Fink et al. [196,197] synthesized polyoxyethylene and polyoxypropylene silicon-containing functional surfactants to better explain the effect of oxygen-containing functional groups on the solubility of silicon-containing polymers in CO_2_. They found that these surfactants have good solubility in CO_2_ and that surfactants with less ethylene oxide addition or less polyoxyethylene propylene chain have better solubility. Low-molecular-weight polyoxyethylene silicon-containing surfactants have a smaller cloud point pressure than high molecular weight polyoxyethylene silicon-containing surfactants. They believe that the polar chain influences solubility more than the molecular weight.

#### 4.2.4. Surfactant Blended Systems

A mixed surfactant is a surfactant system comprising several surfactants that are mixed with other surfactants as agents [198]. The study of water-soluble surfactants showed that the mixture synergistically exhibited better foaming properties than a single surfactant, improving foam stability and reducing crude oil instability. However, some mixed surfactants form crystalline precipitates in aqueous solutions due to the Coulomb interaction between oppositely charged substances [170]. The mixed surfactant system is also severely limited by CO_2_ solubility. Nicolas et al. [173] mixed cationic surfactant F-CAT with anionic surfactant F-OPT, which reduced the static adsorption of surfactant in carbonate reservoirs and significantly improved the apparent viscosity in the presence of oil. The apparent viscosity is mainly determined by the CO_2_ density value, not the temperature, even at different temperatures. It differs from nondense gas foams. Mixed surfactant foams have higher stability and viscosity than single surfactant foams, regardless of the presence of oil. This is mainly due to the close packing of surfactant layers at the gas–liquid interface, which has strong viscoelasticity and leads to higher separation pressure. Zhang et al. [172] studied the synergistic effect of a CO_2_-soluble surfactant mixture comprising an ethoxylamine head group (C_12_NEO_2_) with two coconut oil alkyl tails and a highly ethoxylated nonionic surfactant (C_13_EO_12_). They found that adding C_13_EO_12_ to C_12_NEO_2_ resulted in lower CMC and negative interaction parameter β^σ^. It showed a positive synergistic effect. Although C_13_EO_12_ was difficult to dissolve in CO_2_ alone, the 0.2 wt% surfactant mixture (C_13_EO_12_ ratio is less than 30%) was completely soluble in CO_2_ at 60 °C and 16 MPa to form a transparent single phase. The best surfactant ratio was C_12_NEO_2_:C_13_EO_12_ = 8:2. The volume foam stability and viscosity increased 1.5 times and 2.5 times, respectively. The combined surfactant solution has greater solubility in CO_2_ than a single CO_2_-soluble surfactant and can further improve foam viscosity.

## 5. Conclusions and Future Outlook

Their synergistic effect with CO_2_ can enhance oil recovery more effectively. The application and storage of CO_2_ in oil reservoirs can also effectively alleviate the increase in CO_2_ concentration in the atmosphere and reduce the greenhouse effect. Therefore, CO_2_-soluble surfactants have broad application prospects in oilfield development and have attracted much attention. However, other issues still exist that require additional research.

(1)A CO_2_-soluble surfactant system is divided into CO_2_ + surfactant (+ agents) and CO_2_ + surfactant (+ agents) + water. Notably, adding agents and water aids in increasing CO_2_‘s ability to carry surfactants. Continuous injection is more economical and has great potential. The macro-phase behavior of surfactant, water, and CO_2_ was judged by cloud point pressure in the static experiment to determine the foam stability and solubilization performance of the microemulsion. The foaming performance was then evaluated based on foam volume and foam half-life. The dynamic experimental evaluation method was used to analyze the sealing ability of foam based on the resistance coefficient and residual resistance factor. While CO_2_ can form miscible flooding with crude oil, gas-soluble surfactants can effectively reduce the miscible pressure. Therefore, the interfacial properties can be measured using the hanging drop method to analyze changes in miscible pressure.(2)Snap-off, lamellar division, and leave-behind are the three mechanisms that generate foam in porous media. Foam coalescence is generally due to capillary suction and gas diffusion. Capillary suction is widely considered the main mechanism of foam rupture. Empirical or semi-empirical and mechanical models (population balance models) can analyze foam fluidity. Empirical or semi-empirical models are more targeted, whereas mechanical models are more universal. Although many studies have explored foam performance under different conditions and the improvement of foam mobility models, research on CO_2_-soluble surfactants must be combined with effective on-site cases to provide more valuable information for future development.(3)CO_2_ foam can effectively improve the sweep range and oil displacement efficiency of CO_2_ by reducing the oil–gas mobility ratio, selective plugging, and reducing interfacial tension to change rock wettability. Retention foam can prolong the contact time between CO_2_ and oil, resulting in a better CO_2_ flooding mechanism. However, the study of minimum miscible pressure reduction in the CO_2_–oil system with gas-soluble surfactants remains in the exploratory stage. Many issues need to be addressed. For instance, as the mechanism of agents in the system is not sufficiently evident, improving the interaction mechanism between CO_2_, surfactants, agents, and oil is highly desirable.(4)Surfactant solubility in CO_2_ is the primary parameter for screening CO_2_-soluble surfactants, followed by the analysis of foam properties, thermal stability, and adsorption behavior. At present, only a few CO_2_-philic surfactants exist. Although fluorine- and silicon-containing surfactants have high surface activity, small cohesive energy, and good compatibility with CO_2_, difficult degradation, high toxicity, and high cost seriously limit their field applications. The existing CO_2_-philic hydrocarbon surfactants are affordable, but their solubility in CO_2_ is small, with limited foam performance. Many current studies focus on nonionic surfactants. Anionic, cationic, and zwitterionic surfactants, as well as two or more mixed surfactants, have also been studied, although their overall foaming performance requires further improvement. Overall, enhancing the foam performance of nonionic surfactants or improving the solubility of ionic surfactants in CO_2_ will be effective in realizing improved oil recovery efficiency.(5)This article explores the application of CO_2_-soluble surfactants in tight reservoirs with low permeability and high reservoir heterogeneity. Advanced screening and simulation techniques are introduced to more accurately predict the behavior of surfactants in reservoirs, reducing the time and cost of experimental stages.(6)Future research should focus on developing new CO_2_-soluble surfactants that address current limitations while enhancing performance in challenging reservoir conditions. One promising direction is the development of environmentally friendly and biodegradable surfactants. With increasing environmental regulations and the need for sustainable oil recovery methods, the use of green chemistry to design surfactants that are both highly effective and environmentally benign will become critical. Additionally, fluorinated surfactants and siloxane-based surfactants have shown potential for improving solubility and stability in supercritical CO_2_, especially under high temperature and salinity conditions. Research could also explore the development of nanoparticle-enhanced surfactants, which combine the stability of nanoparticles with surfactant-based foam systems to further increase foam stability and extend the duration of mobility control. Another promising direction is the customization of surfactant molecular structures to target specific reservoir characteristics. For example, surfactants with branched or double-tail hydrophobic groups could be designed to improve solubility in CO_2_ while introducing functional groups that enhance interactions with the formation of water could lead to more stable foam films in highly heterogeneous or low-permeability reservoirs.

## Figures and Tables

**Figure 1 molecules-29-05411-f001:**
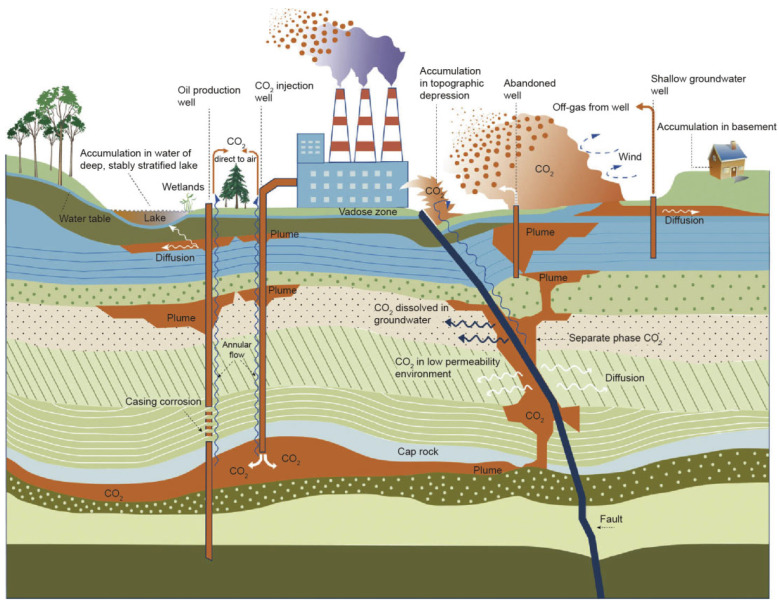
Mechanism and effect of CO_2_ leakage in geological buried sites [7].

**Figure 2 molecules-29-05411-f002:**
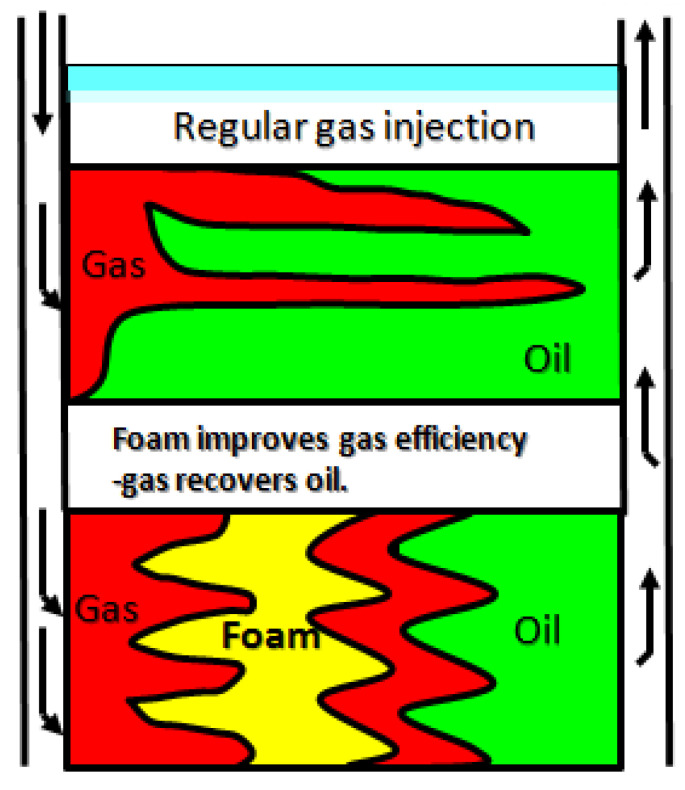
Improvement of gas transport in porous media by foam.

**Figure 3 molecules-29-05411-f003:**
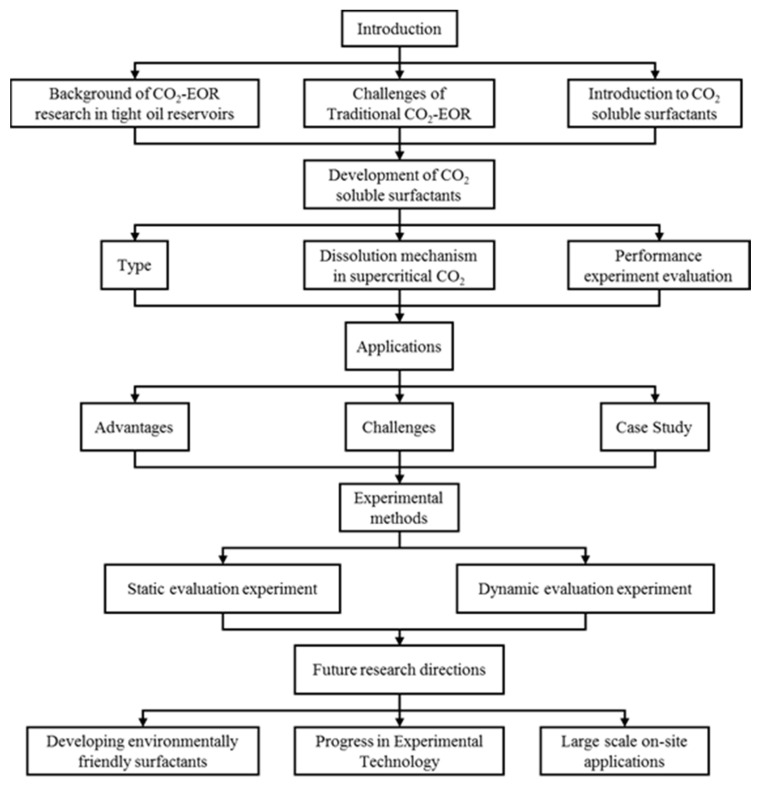
Scope of the review.

**Figure 4 molecules-29-05411-f004:**
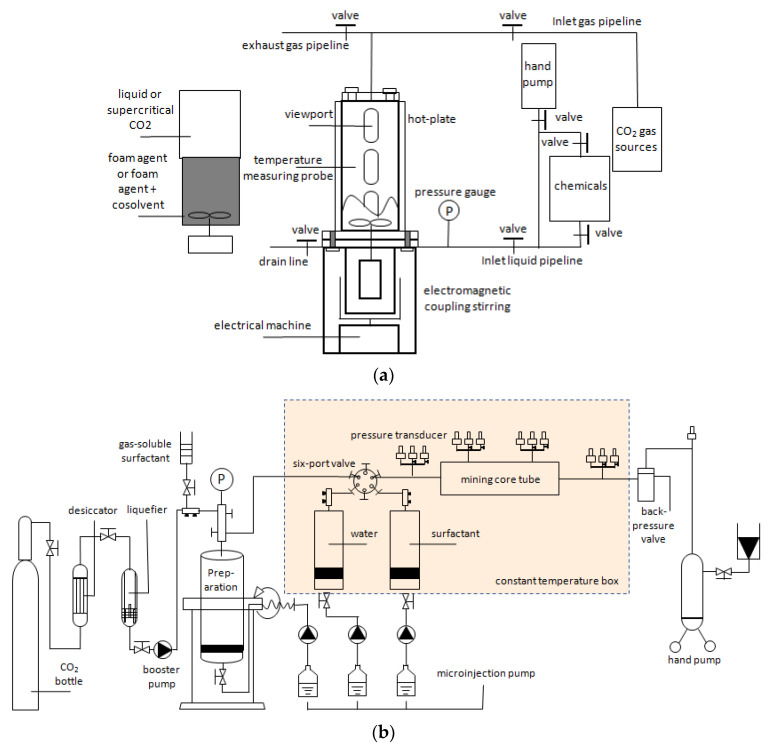
Laboratory experimental study of CO_2_ foam system: (**a**) static experiment [34] and (**b**) dynamic experiment [33].

**Figure 5 molecules-29-05411-f005:**
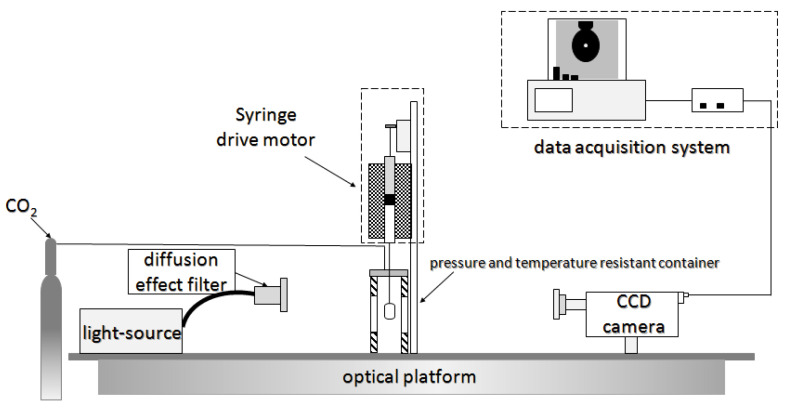
Interface tension experimental device [33].

**Figure 6 molecules-29-05411-f006:**
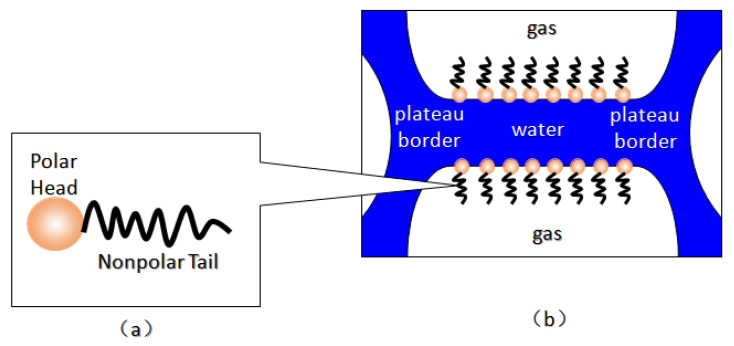
Foam structure in porous media ((**a**) Surface active molecules; (**b**) lamellar structure) [62].

**Figure 7 molecules-29-05411-f007:**
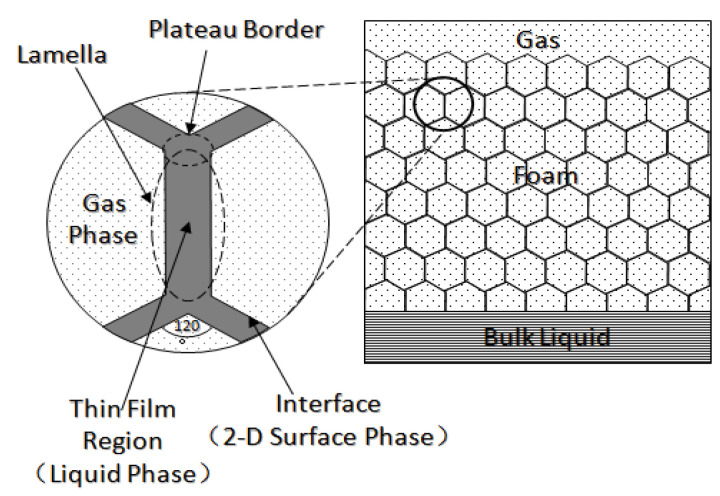
Generalized foam system [61].

**Figure 8 molecules-29-05411-f008:**
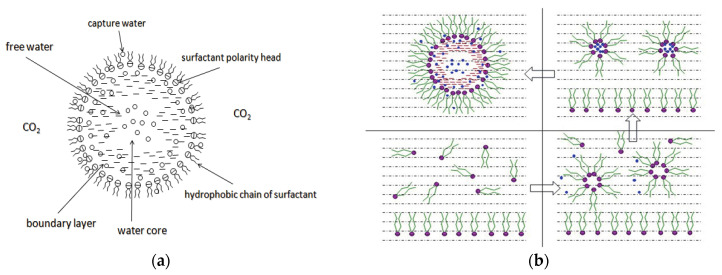
(**a**) Reverse micelle structure in supercritical CO_2_ microemulsion; (**b**) process of surfactant formation solubilization micelle in supercritical CO_2_ [34].

**Figure 9 molecules-29-05411-f009:**
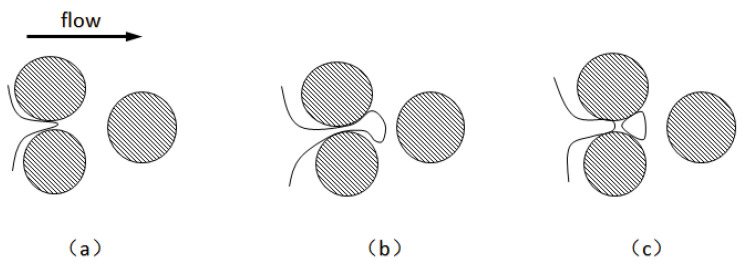
Foam snap-off process ((**a**) into pores, (**b**) through pores, (**c**) after passing through pores) [90].

**Figure 10 molecules-29-05411-f010:**
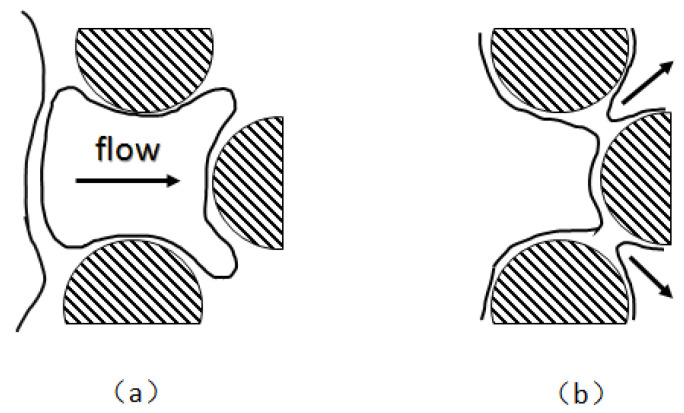
Lamella division diagram of foam ((**a**) before lamella division and (**b**) after lamella division) [90].

**Figure 11 molecules-29-05411-f011:**
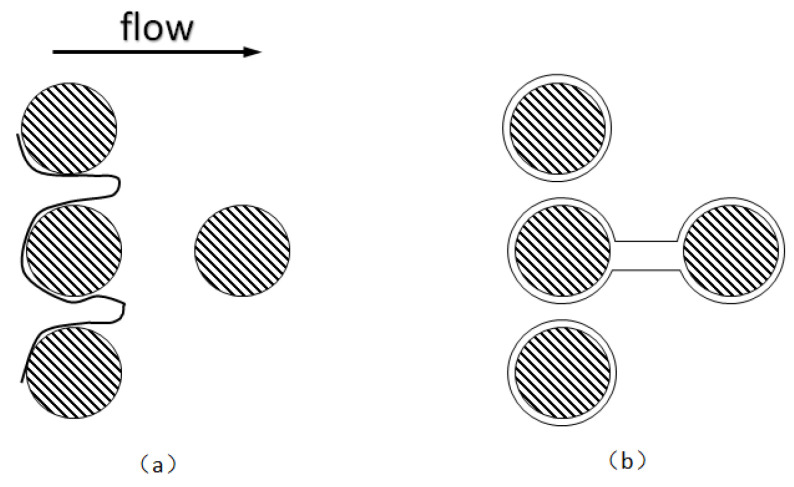
Schematic diagram of leave-behind ((**a**) before the foam through and (**b**) after the foam through) [92].

**Figure 12 molecules-29-05411-f012:**
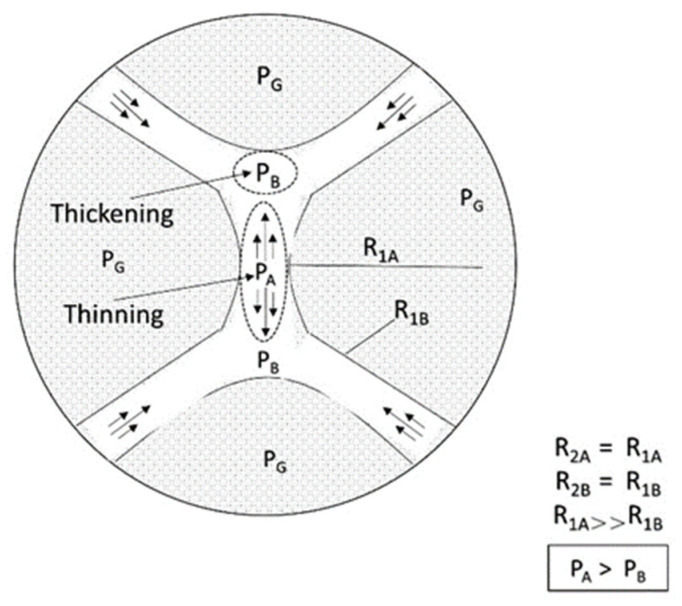
Pressure difference between the center and the edge of the lamella [51].

**Figure 13 molecules-29-05411-f013:**
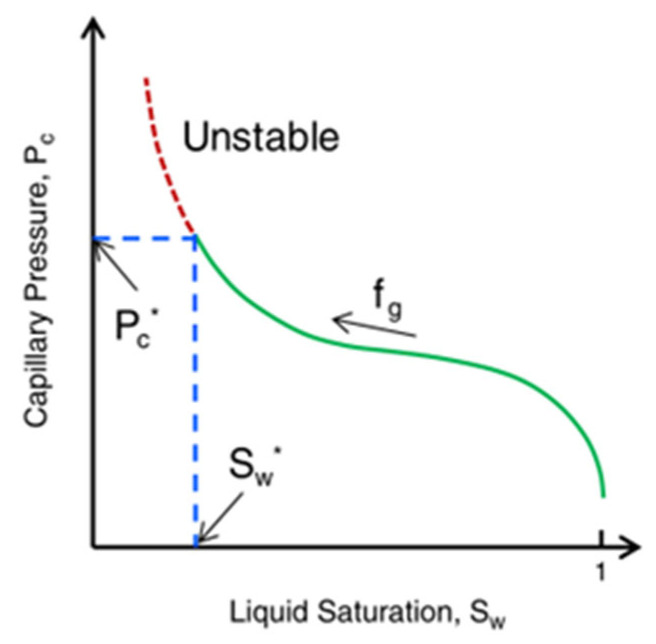
Capillary pressure curve (f_g_ is gas fraction flow) [63].

**Figure 14 molecules-29-05411-f014:**
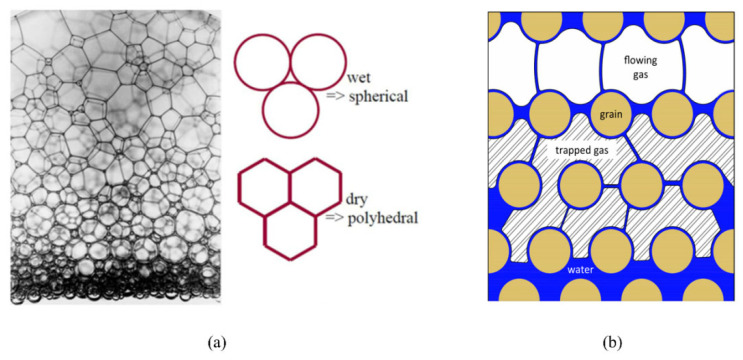
Forms of foam existence ((**a**) bulk foam [94] and (**b**) foam in porous media [119]).

**Figure 15 molecules-29-05411-f015:**
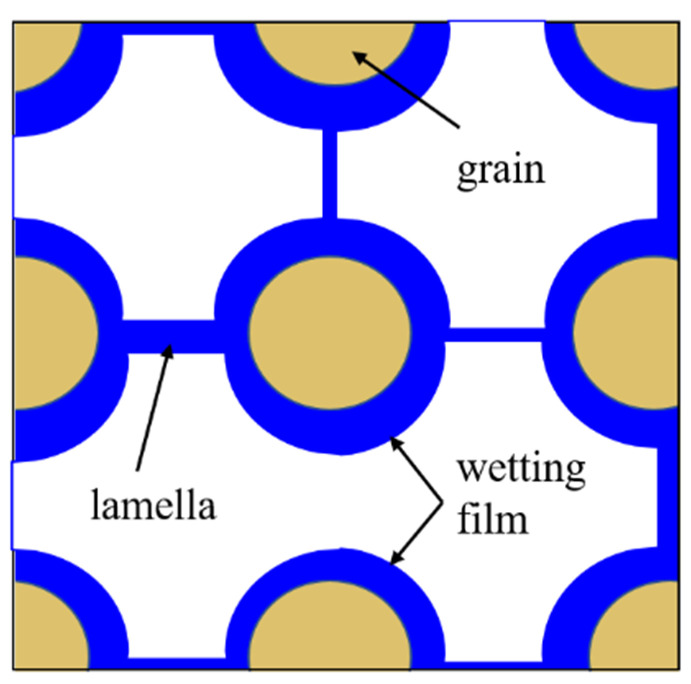
Gas and liquid distribution diagram of strong foam in porous media [121].

**Figure 16 molecules-29-05411-f016:**
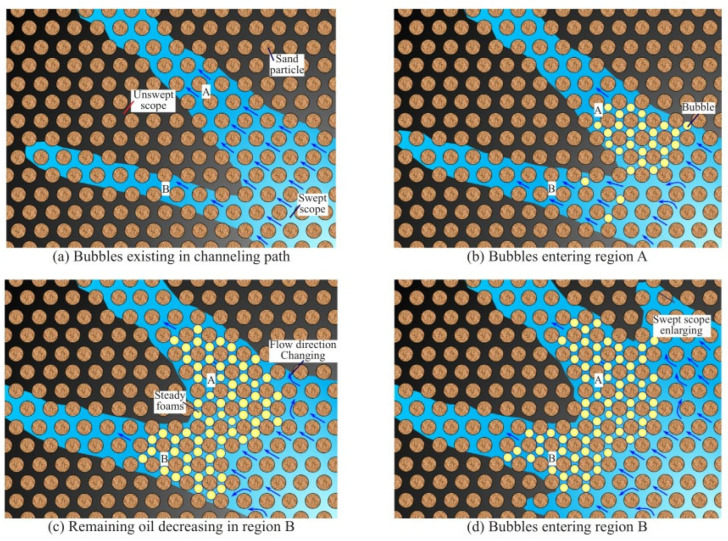
Increased sweep range by foam plugging ((**a**) bubbles existing in channeling path, (**b**) bubbles entering region A, (**c**) remaining oil decreasing in region B, (**d**) bubbles entering region B) [149].

**Figure 17 molecules-29-05411-f017:**
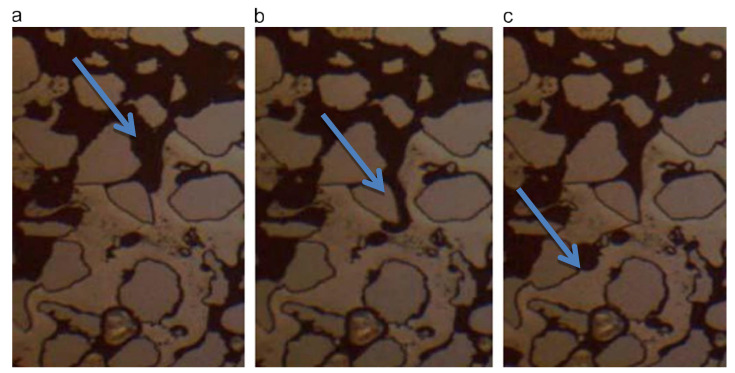
Changes in rock wettability facilitate the displacement of residual oil ((**a**) After water flooding, the residual oil is prevented from passing through pores by capillary forces, (**b**) the residual oil is mobilized when it comes into contact with an alkaline surfactant, (**c**) the mobilized oil mass is pulled into the flow channel to form a microscopic oil droplet that is eventually washed away by water flow) [153].

**Table 3 molecules-29-05411-t003:** Summary of common CO_2_-soluble surfactants.

Surfactants	Trade Name	Chemical Expression	CAS
Nonionic surfactants
2-(2-[4-(1,1,3,3-Tetramethylbutyl)phenoxy]ethoxy)ethanol	Dow Triton X 100 (x = 10), BASF Lutensol OP 10 (x = 10), Huntsman SURFONIC^®^ OP-100 (x = 10), OP-120 (x = 12)	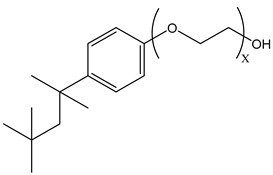	9036-19-5
Nonylphenol branched ethoxylated	Huntsman SURFONIC^®^ N-120, N-150, N-200, N-300, N-400, x = 12, 15, 20, 30, 40.	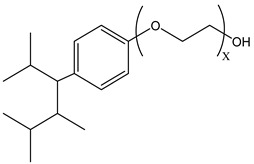	127087-87-0
	Dow Tergitol NP9,12,15 (x = 9, 12, 15)	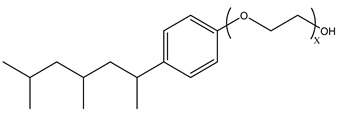	127087-87-0
	Stepan Cedepal CO 630, 710, x = 10 and 10.5	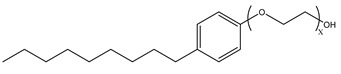	127087-87-0
Tristyryl phenol ethoxylated	Huntsman XOF-501	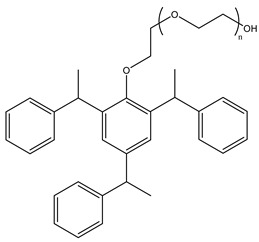	99734-09-5
Polyethylene glycol trimethylnonyl ether	Dow trimethylnonyl Tergitol TMN 6	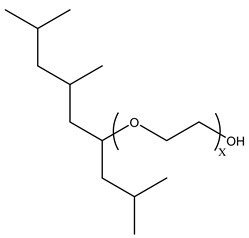	60828-78-6
Ethoxylated isodecyl alcohol	BASF Lutensol XP 70	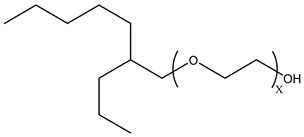	61827-42-7
C_12_-C_14_ fatty alcohols ethoxylated	BASF Lutensol TO 8, 10	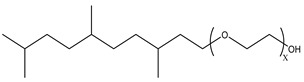	68439-50-9
GENAPOL(R) X-080	Huntsman SURFONIC^®^ TDA-8, 9	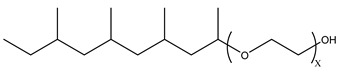	9043-30-5
Alkyl-(C_10_-C_14_) alcohol, ethoxylated	Huntsman SURFONIC^®^ L12-8; BASF Lutensol AO8, AO11	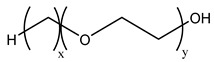	66455-15-0
Polyethylene glycol monolaurate	Sigma Aldrich PEG monolaurate 600	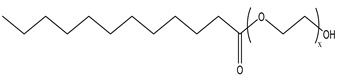	9004-81-3
polyoxyethylene 20 sorbitan monooleate	Tween 80	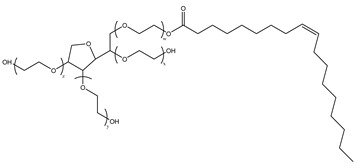	9005-65-6
N,N’,N’-polyoxyethylene (10)-N-tallow-1,3-diaminopropane	Ethoduomeen T/13	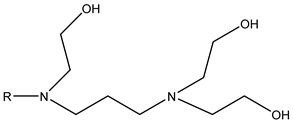	61790-85-0
Coco alkyldimethylamines	Armeen DMCD	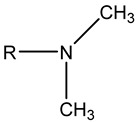	61788-93-0
Propoxylated and ethoxylated dodecanol	-	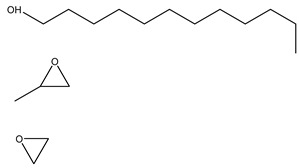	68238-81-3
Oxirane, methyl-, polymer with oxirane, mono(2-ethylhexyl) ether	-	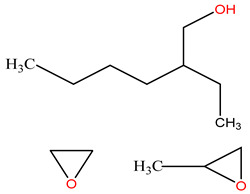	64366-70-7
Anionic surfactants
Dioctyl sulfosuccinate sodium salt	AOT	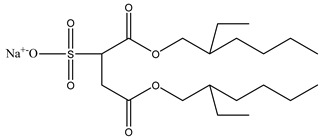	577-11-7
Sodium dodecyl sulfate	SDS	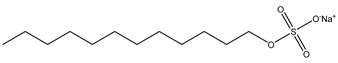	151-21-3
Cationic surfactants
N,N’,N’-Trimethyl-N-(tallowalkyl)-1,3-propanediamine	DTM	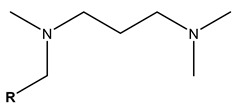	111-33-1
Ethoxylated cocoamines	Ethomeen C12	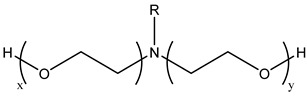	61791-14-8
Zwitterionic surfactants
Cocoyl amide propyldimethyl glycine	CAB-35	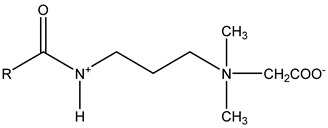	86438-79-1
LauroylaMide propylbetaine	-	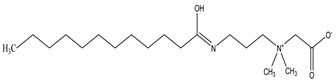	4292-10-8
N-Dodecyl-N,N-dimethyl-3-ammonio-1-propanesulfonate	LDMAA	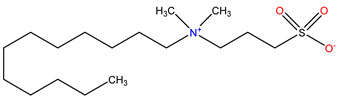	14933-08-5
Fluorine- and nitrogen-containing surfactants
Perfluoropolyether ammonium carbonate	-	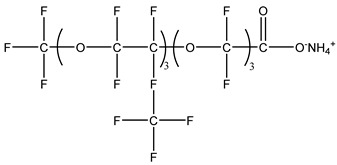	-
Double-tailed fluorocarbon sulfate	-	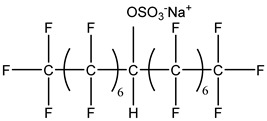	-
Double-tailed fluorocarbon-hydrocarbon mixed sulfate	-	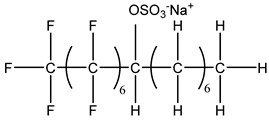	-
Fluorine-containing dialkyl phosphate	-	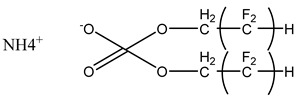	-
Fluorine-containing AOT homolog	-	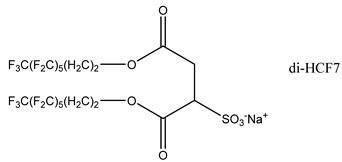 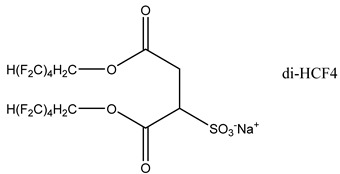 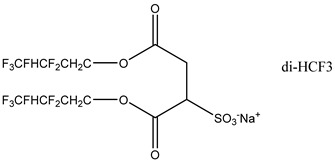	-
Oligosiloxane	-	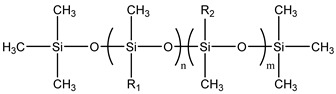	-
Functional silicone	-	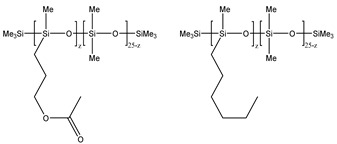	-

**Table 4 molecules-29-05411-t004:** CO_2_-soluble surfactant blend systems.

Surfactant	Remarks	Reference
Double-tailed anion: sodium dihexyl sulfosuccinate (SDHS). Unbalanced-tail (different tail lengths) cation: benzethonium chloride (BCl)	Microemulsions can be formed without adding alcohol. Compared with the use of anionic surfactants alone, the mixture exhibited a higher critical microemulsion concentration. Under optimal microemulsion conditions, the mixed anionic–cationic surfactant system solubilizes more oil than the anionic surfactant alone.	[170]
Anionic: F-OPT. Cationic: F-CAT	At high temperature (80 °C), high salinity (160 g/L TDS), high hardness (R+ = 0.3), and pressure (120 bar) conditions can be effectively dissolved. The mixed system can reduce the adsorption of carbonate powder. Under supercritical CO_2_ (40 °C/120 bar) conditions, the half-life of volume foam in carbonate minerals (99% alcite) achieves a low static state of 24 h.	[173]
Anion: Sodium bis(1H,1H,2H,2H-heptadecafluorodecyl)-2-sulfosuccinate (8FS(EO)_2_); fluorocarbon–hydrocarbon hybrid anionic surfactants (FC6-HCn)	In the presence of excess water, the mixed surfactant can prevent the conversion of the microemulsion to the liquid crystal phase. At the same time, it was found that the micro-separation of 8FS(EO)_2_ and FC6-HCn formed a loose molecular accumulation, which enhanced the stability of the mixed microemulsion and the area occupied by each surfactant molecule.	[171]
Two ethoxylated amine headgroups with cocoalkyl tails (C_12_NEO_2_) and nonionic surfactant with high degree ethoxylation (C_13_EO_12_)	There is a positive synergy between the two surfactants, which can effectively improve the foam stability. When the C_13_EO_12_ ratio is less than 30%, the cloud point pressure increment will be less than 20%. At the optimum ratio, the apparent viscosity of foam increases by about 2.5 times.	[172]

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
