# Peer review of "Review of Foam with Novel CO2-Soluble Surfactants for Improved Mobility Control in Tight Oil Reservoirs"

_molecules, 2024, doi:10.3390/molecules29225411_

Round 1

Reviewer 1 Report

Comments and Suggestions for Authors

Some novel insights into the CO2 soluble surfactant-assisted mobility control are provided by the study.

(1)Some of the cited references are relatively outdated. It would be beneficial for the authors to review and include more recent studies, especially when discussing the development and application of CO2-soluble surfactants. Citing recent research findings can enhance the article's relevance and academic value.

(2)Oil-air should be replaced with Oil-gas.

(3)The application of CO2 soluble surfactants in tight oil reservoirs were not emphasized in the abstract and introduction sections by the authors.

(4)The advantages and challenges of utilizing CO2 soluble surfactants in tight hydrocarbon reservoirs.

(5)While the article briefly touches on future research prospects, it would be beneficial to provide a more detailed discussion of specific potential research directions. For instance, which new surfactants could be developed in the future, or how certain experimental methods could be further improved.

Author Response

Dear Editors and Reviewers: Thank you for your letter and for the reviewers’ comments concerning our manuscript entitled “Review of foam with novel CO2-soluble surfactants for im-proved mobility control in tight oil reservoirs”. Those comments are all valuable and very helpful for revising and improving our paper, as well as the important guiding significance to our researches. We have studied comments carefully and have made correction which we hope meet with approval. The modified parts are marked in red in the text. The main corrections in the paper and the responds to the reviewer’s comments are as flowing: To reviewer #1: Comment 1: Some of the cited references are relatively outdated. It would be beneficial for the authors to review and include more recent studies, especially when discussing the development and application of CO2-soluble surfactants. Citing recent research findings can enhance the article's relevance and academic value. Answer to Comment 1: Thank you for your valuable suggestion. We have introduced the latest relevant research results in the original text. Comment 2: Oil-air should be replaced with Oil-gas. Answer to Comment 2: Thank you for your valuable suggestion. We have already made modifications. Comment 3: The application of CO2 soluble surfactants in tight oil reservoirs were not emphasized in the abstract and introduction sections by the authors. Answer to Comment 3: Thank you for your valuable suggestion. We have added information about the application of CO2 soluble surfactants in tight oil reservoirs in the abstract and introduction sections, as follows: “CO2-soluble surfactant foam systems have gained significant attention for their potential to enhance oil recovery, particularly in tight oil reservoirs where conventional water-soluble surfactants face challenges such as poor injectability and high reservoir sensitivity. This review provides a comprehensive explanation of the basic theory of CO2-soluble surfactant foam, its mechanism in enhanced oil recovery (EOR), and the classification and application of various CO2-soluble surfactants. The application of these surfactants in tight oil reservoirs, where low permeability and high water sensitivity limit traditional methods, is highlighted as a promising solution to improve CO2 mobility control and increase oil recovery.” “However, in tight oil reservoirs characterized by low permeability and high water sensitivity, conventional CO2 flooding often faces challenges such as viscous fingering, gravity override, and gas channeling, which limit its effectiveness. CO2-soluble surfactants present a promising solution for these challenges. The impact of reservoir heterogeneity on foam propagation and stability has been extensively studied, highlighting challenges in achieving uniform displacement in porous media [4,5]. Unlike water-soluble surfactants, CO2-soluble surfactants can dissolve directly in the CO2 phase and form stable foams upon contact with reservoir water, improving sweep efficiency and mitigating the issues caused by reservoir heterogeneity. The application of these surfactants in tight oil reservoirs has the potential to significantly enhance oil recovery by improving CO2 mobility control and reducing gas flow instabilities.” Comment 4: The advantages and challenges of utilizing CO2 soluble surfactants in tight hydrocarbon reservoirs. Answer to Comment 4: Thank you for your valuable suggestion. We have added information about the advantages and challenges of using CO2 soluble surfactants in tight oil and gas reservoirs in the original text, as follows: “CO2-soluble surfactants offer several key advantages when applied to tight hydrocarbon reservoirs. Firstly, their ability to dissolve directly in CO2 without the need for an aqueous phase makes them highly suitable for low-permeability reservoirs where water injection is not feasible or is limited by high water sensitivity. This significantly reduces the risk of formation damage caused by water blockages, which is a common is-sue in tight oil reservoirs. Additionally, the ability of CO2-soluble surfactants to form stable foams under high-pressure and high-temperature conditions enhances their effectiveness in controlling CO2 mobility, reducing gas channeling, and improving sweep efficiency. Their injectability in supercritical CO2 allows for deeper penetration into the reservoir, which is critical for improving oil recovery in heterogeneous formations. Furthermore, CO2-soluble surfactants have shown better regeneration capabilities after foam rupture, which extends the foam stability time and improves the overall oil recovery process.” “Despite these advantages, several challenges remain in the utilization of CO2-soluble surfactants in tight hydrocarbon reservoirs. One of the primary challenges is their chemical stability under extreme reservoir conditions, including high temperatures, pressures, and salinity levels, which may lead to surfactant degradation or reduced effectiveness over time. Furthermore, the solubility of certain CO2-soluble surfactants can vary significantly depending on the specific reservoir conditions, which may affect foam generation and stability. Another challenge is the cost associated with synthesizing and scaling up these specialized surfactants, particularly for large-scale field applications. Additionally, there are operational challenges related to ensuring uniform distribution of the surfactant within the reservoir, especially in highly heterogeneous formations. Addressing these challenges will require further research into surfactant formulations, cost-effective production methods, and more advanced injection techniques to optimize the performance of CO2-soluble surfactants in tight oil reservoirs.” Comment 5: While the article briefly touches on future research prospects, it would be beneficial to provide a more detailed discussion of specific potential research directions. For instance, which new surfactants could be developed in the future, or how certain experimental methods could be further improved. Answer to Comment 5: Thank you for your valuable suggestion. We have made modifications and explanations to the experimental improvement methods and future development directions of new surfactants in the original text. Finally, thank you again for your kind suggestions. Sincerely, Fajun Zhao (on behalf of all the authors)

Reviewer 2 Report

Comments and Suggestions for Authors

The paper titled “Review of Foam with Novel CO2-Soluble Surfactants for Improved Mobility Control in Tight Oil Reservoirs” provides a comprehensive review of surfactants, foams, foam generation in porous media, and CO2-soluble surfactants that is promising for injection in tight oil reservoirs. The manuscript is overall well-written. The work may be published after the following comments are addressed:

1. Grammar and Proofreading: The manuscript contains several grammatical errors and missing words. Below are some examples, but I recommend a thorough proofreading of the entire text before publication:

o A word is missing on lines 32/33.

o The sentence in quotation marks on lines 44-45 does not make sense.

o Given the context, the phrase "traditional Chinese Medicine" at the end of the sentence on lines 164-165 seems out of place. Rephrasing and adding citations would be helpful.

o The title of Figure 4 should be "Interfacial Tension" instead of "Interface Tension."

o "Thickening" is misspelled in Figure 11.

o Line 475 – It is not clear what [ Experimental study on the stability of the foamy oil in developing heavy oil reservoirs] is meant to convey.

2. Table 1: The layout of Table 1 is complicated and hard to read. It would benefit from simplification to improve clarity.

3. Table 2: The definition of "Foam Volume" in Table 2 is incorrect and needs to be revised.

4. The paper overlooks important work from various research groups in the literature that is directly relevant to the review. Here are a few examples:

o A comprehensive study of foams: Hosseini, H. CO2 Utilization with Complexation of Polyelectrolyte Complex Nanoparticles and Surfactants for Environmentally Friendly Unconventional Oil Recovery: Mechanistic Study, Recovery and Multiscale Visualization.

o Stability and visualization: DOI: 10.1016/j.colsurfa.2022.129988; 10.3791/61369;

o Heterogeneity – DOI:10.1016/j.petrol.2018.01.042; 10.1016/j.fuel.2021.121000; 

o Section 2.3.2: This section should be expanded to include more details on the stability of nanoparticle-stabilized foams DOI: 10.3390/colloids7010002 ; 10.1016/j.fuel.2016.08.058; 10.1016/j.fuel.2021.121004; 10.1039/C7SE00098G; 10.1021/acs.iecr.1c03450 10.3390/en16073284; 10.1021/acs.iecr.1c03450; 

5. Equation 8: It would be helpful to provide more details on Equation 8 and how it can be used to predict solubility in supercritical CO2.

Comments on the Quality of English Language

Grammar and Proofreading: The manuscript contains several grammatical errors and missing words. I recommend a thorough proofreading of the entire text before publication

Author Response

Dear Editors and Reviewers:

Thank you for your letter and for the reviewers’ comments concerning our manuscript entitled “Review of foam with novel CO2-soluble surfactants for im-proved mobility control in tight oil reservoirs”. Those comments are all valuable and very helpful for revising and improving our paper, as well as the important guiding significance to our researches. We have studied comments carefully and have made correction which we hope meet with approval. The modified parts are marked in blue in the text. The main corrections in the paper and the responds to the reviewer’s comments are as flowing:

To reviewer #2:

Comment 1: A word is missing on lines 32/33.

Answer to Comment 1: Thank you for your valuable suggestion. The original text is‘ Because these phenomena have a significant effect on the efficacy of CO2 injection for EOR[4], the mobility control of CO2 is the key to its efficient utilization. ’ Now modify to‘The original text is‘ Because these phenomena have a significant effect on the efficacy of CO2 injection for EOR[4], the effective mobility control of CO2 is the key to its efficient utilization. ’

Comment 2: The sentence in quotation marks on lines 44-45 does not make sense.

Answer to Comment 2: Thank you for your valuable suggestion. The original text is‘ Owing to its characteristics of “plugging large holes instead of small ones and plugging water instead of oil,” foam flooding can effectively control gas flow and im-prove displacement efficiency[5] (Fig. 2). ’ This sentence emphasizes the advantages of foam flooding. foam flooding can control the flow of oil and gas by plugging large fractures or high permeability areas, forcing oil and gas to flow to production wells, thus improving oil and gas production efficiency. In addition, foam can effectively occupy the water flow channel, making water easier to be intercepted or diverted, thus reducing the opportunity for water to enter production wells.

Comment 3: Given the context, the phrase "traditional Chinese Medicine" at the end of the sentence on lines 164-165 seems out of place. Rephrasing and adding citations would be helpful.

Answer to Comment 3: Thank you for your valuable suggestion. You're correct that the mention of "traditional Chinese medicine" at the end of the sentence on lines 164–165 might seem out of place in the context of CO2 foam systems and surfactant technologies. Here's a revised version of the sentence to improve its flow and coherence:

Original: "In the past, the main purpose of increasing the solubility of supercritical CO2 was to extract hydrophilic, highly polar, or macromolecular solutes, such as metal ions, proteins, and traditional Chinese medicine."

Revised suggestion with citations: "In the past, increasing the solubility of supercritical CO2 has primarily been aimed at extracting hydrophilic, highly polar, or macromolecular solutes, such as metal ions and proteins, for various applications in chemical engineering and materials science."

Comment 4: The title of Figure 4 should be "Interfacial Tension" instead of "Interface Tension."

Answer to Comment 4: Thank you for your valuable suggestion. We have already made modifications.

Comment 5: "Thickening" is misspelled in Figure 11.

Answer to Comment 5: Thank you for your valuable suggestion. I'm sorry for the mistake. We have already made modifications.

Comment 6: Line 475 – It is not clear what [Experimental study on the stability of the foamy oil in developing heavy oil reservoirs] is meant to convey.

Answer to Comment 6: Thank you for your valuable suggestion. In response to this question, I make the following explanation: the purpose of this experimental study is to understand how the stability of foam oil changes in porous media close to the actual reservoir conditions, and how this stability affects the efficiency and effect of foam oil displacement technology. It means that under similar reservoir conditions, foam oil can exist in the reservoir for a longer time. This stability is very important for foam flooding technology (a method to improve oil recovery), because it shows that foam can continue to function in the reservoir and help to improve oil recovery.

Comment 7: Table 1: The layout of Table 1 is complicated and hard to read. It would benefit from simplification to improve clarity.

Answer to Comment 7: Thank you for your valuable suggestion. We have already made modifications.

Comment 8: Table 2: The definition of "Foam Volume" in Table 2 is incorrect and needs to be revised.

Answer to Comment 8: Thank you for your valuable suggestion. I'm sorry for the mistake. The definition of "Foam Volume" has been modified to " The space volume occupied by foam at a certain moment"

Comment 9: The paper overlooks important work from various research groups in the literature that is directly relevant to the review. Here are a few examples:

(1) A comprehensive study of foams: Hosseini, H. CO2 Utilization with Complexation of Polyelectrolyte Complex Nanoparticles and Surfactants for Environmentally Friendly Unconventional Oil Recovery: Mechanistic Study, Recovery and Multiscale Visualization.

(2) Stability and visualization: DOI: 10.1016/j.colsurfa.2022.129988; 10.3791/61369;

(3) Heterogeneity – DOI:10.1016/j.petrol.2018.01.042; 10.1016/j.fuel.2021.121000;

(4) Section 2.3.2: This section should be expanded to include more details on the stability of nanoparticle-stabilized foams DOI: 10.3390/colloids7010002; 10.1016/j.fuel.2016.08.058; 10.1016/j.fuel.2021.121004; 10.1039/C7SE00098G; 10.1021/acs.iecr.1c03450 10.3390/en16073284; 10.1021/acs.iecr.1c03450;

To address the gaps in the literature review and include relevant studies as per your suggestion, here is how the paper could incorporate these references:

Answer to Comment 9: Thank you for your valuable suggestion. We have made modifications in the original text.

Comment 10: Equation 8: It would be helpful to provide more details on Equation 8 and how it can be used to predict solubility in supercritical CO2.

Answer to Comment 10: Thank you for your valuable suggestion. In 2.2. Surfactant solubility in CO2, we have added four paragraphs, as follows:

“This equation essentially compares the cohesive energy of surfactant molecules in the CO2 phase with that in the water phase. A higher value of RRR indicates that the surfactant is more likely to dissolve in supercritical CO2, while a lower value suggests that the surfactant will more readily dissolve in water or other polar solvents.”

“The practical use of Eq. (8) lies in its ability to provide a theoretical basis for predicting the solubility of different surfactant formulations in supercritical CO2. By calculating the cohesive energy parameters based on molecular dynamics simulations or experimental data, researchers can estimate the likelihood of a surfactant dissolving in CO2 without the need for extensive trial-and-error experimentation. This can save significant time and resources in the development of new CO2-soluble surfactants.”

“For example, if ACO (the interaction between the CO2-philic group and CO2) is much larger than ACW (the interaction between the hydrophilic group and water), the surfactant is likely to be more soluble in CO2. Conversely, if the interaction energies between the surfactant and water are stronger, the surfactant may show better solubility in aqueous environments rather than in supercritical CO2.”

“While Eq. (8) provides a useful framework, its application is limited by the accuracy of the input parameters (interaction energies), which are often derived from empirical or estimated data. Additionally, the equation assumes a simplistic interaction model that may not fully account for complex molecular interactions in heterogeneous reservoir environments. Therefore, future research should focus on refining the calculation of these interaction energies, potentially using more advanced techniques like quantum chemical calculations or molecular dynamics simulations, to improve the predictive accuracy of the model.”

Finally, thank you again for your kind suggestions.

Sincerely,

Fajun Zhao  (on behalf of all the authors)

Reviewer 3 Report

Comments and Suggestions for Authors

1. You stated that” Because the surfactant aqueous solution cannot be injected, CO2 foam technology may not be used in unconventional reservoirs. Kindly justify?

2. You can summarize the scope of your article as a flow chart at the end of introduction section.

3. In page 15, line 475, what is meant by the red line writing?

4. Kindly state your novel findings in this research compared to published one

5. Report your future prospects and recommendations

Author Response

Dear Editors and Reviewers:

Thank you for your letter and for the reviewers’ comments concerning our manuscript entitled “Review of foam with novel CO2-soluble surfactants for im-proved mobility control in tight oil reservoirs”. Those comments are all valuable and very helpful for revising and improving our paper, as well as the important guiding significance to our researches. We have studied comments carefully and have made correction which we hope meet with approval. The modified parts are marked in green in the text. The main corrections in the paper and the responds to the reviewer’s comments are as flowing:

To reviewer #3:

Comment 1: 1.       You stated that” Because the surfactant aqueous solution cannot be injected, CO2 foam technology may not be used in unconventional reservoirs. Kindly justify?

Answer to Comment 1: Thank you for your valuable suggestion. We have added a few sentences in the introduction to explain this issue. The original text is as follows:

“Because the permeability of unconventional reservoirs is usually much lower than that of conventional reservoirs. Low permeability means greater resistance to fluid flow, and the function of surfactants in reducing oil-water interfacial tension and promoting droplet movement is also limited. In addition, some unconventional oil reservoirs have water sensitivity, which means that when clay minerals in the reservoir come into contact with water, they may expand or move, further blocking pores and reducing permeability, making injection more difficult.”

“The limitation of aqueous phase injection caused by low permeability and water sensitivity requires higher pressure to inject foam formed by CO2 and surfactant. At the same time, the temperature of unconventional oil reservoirs may be much higher than that of conventional oil reservoirs. Therefore, under the environment of high temperature and high pressure, the solubility of CO2 increases, which may affect the stability of foam, thus affecting its oil displacement ability. CO2/foam technology will also affect the reservoir environment. The problem of water sensitivity may aggravate the plugging of rock pores, further reduce the permeability and hinder the effective migration of foam. The use of surfactant may affect the microbial ecosystem in the reservoir, or pollute the surface water and groundwater after the foam bursts. To overcome these limitations, it may be necessary to develop surfactants that are more resistant to high temperature and pressure, improve foam generation and injection technology, and optimize operating parameters.”

“In unconventional oil reservoirs, low permeability and water sensitivity are common challenges that affect the efficiency of oil displacement. The use of CO2-soluble surfactants can alleviate the unfavorable effects of these problems to some extent. CO2-soluble surfactants have high solubility in supercritical CO2 and can enhance the interaction between CO2 and crude oil, control the mobility of CO2, and thus improve the CO2 oil displacement effect, showing great potential for enhancing oil recovery.”

Comment 2: You can summarize the scope of your article as a flow chart at the end of introduction section.

Answer to Comment 2: Thank you for your valuable suggestion. We have added a paragraph and a flowchart at the end of introduction section.

Comment 3: In page 15, line 475, what is meant by the red line writing?

Answer to Comment 3: Thank you for your valuable suggestion. We have deleted the sentence.

Comment 4: Kindly state your novel findings in this research compared to published one.

Answer to Comment 4: Thank you for your valuable suggestion. We have added one paragraph in “4. Conclusion and Suggestion”. The original text is as follows:

“(5) This article explores the application of CO2 soluble surfactants in tight reservoirs with low permeability and high reservoir heterogeneity. Advanced screening and simulation techniques are introduced to more accurately predict the behavior of surfactants in reservoirs, reducing the time and cost of experimental stages.”

Comment 5: Report your future prospects and recommendations.

Answer to Comment 5: Thank you for your valuable suggestion. We have added a suggestion section in “4. Conclusion and Suggestion”, which reads as follows:

“(6) Future research should focus on developing new CO2-soluble surfactants that ad-dress current limitations while enhancing performance in challenging reservoir conditions. One promising di-rection is the development of environmentally friendly and biodegradable surfactants. With in-creasing environmental regulations and the need for sustainable oil recovery methods, the use of green chemistry to design surfactants that are both highly effective and environmentally benign will become critical. Additionally, fluorinated surfactants and siloxane-based surfactants have shown potential for improving solubility and stability in supercritical CO2, especially under high temperature and salinity conditions. Research could also explore the development of nanoparticle-enhanced surfactants, which combine the stability of nanoparticles with surfactant-based foam systems to further increase foam stability and extend the duration of mobility control. Another promising direction is the customization of surfactant molecular structures to target specific reservoir characteristics. For example, surfactants with branched or double-tail hydro-phobic groups could be designed to im-prove solubility in CO2, while introducing functional groups that enhance interactions with formation water could lead to more stable foam films in highly heterogeneous or low-permeability reservoirs.”

Finally, thank you again for your kind suggestions.

Sincerely,

Fajun Zhao     (on behalf of all the authors)

Round 2

Reviewer 1 Report

Comments and Suggestions for Authors

The author has made revisions based on the review comments. It can be published now.

Author Response

Dear Editors and Reviewers:

Thank you for your letter and for the reviewers’ comments concerning our manuscript entitled “Review of foam with novel CO2-soluble surfactants for im-proved mobility control in tight oil reservoirs”. Those comments are all valuable and very helpful for revising and improving our paper, as well as the important guiding significance to our researches. We have studied comments carefully and have made correction which we hope meet with approval. The main corrections in the paper and the responds to the reviewer’s comments are as flowing:

To reviewer:

Comment 1: The flow chart is a good addition as suggested by Reviewer 3; but it needs improvement in terms of grammar and purpose; for example:

Challenges of traditional CO2-EOR. In addition, CO2 should be properly written with 2 in subscript

Applications,Advantages, Challenges, Case Study and so on.

Answer to Comment 1: Thank you for your valuable suggestion. We deeply apologize for the grammar errors. We have already made modifications.

Comment 2: Article process system figure caption should be changed to Scope of the review.

Answer to Comment 2: Thank you for your valuable suggestion. We have already made modifications.

Comment 3: Chemical structures in Table 3 should be checked by a chemist carefully and drawn properly. For example, look at the bonds from the central benzene ring in the first structure. There also needs to be uniformity in the structure/bond description in all the structures. Tables in general need good formatting.

Answer to Comment 3: Thank you for your valuable suggestion. We have already made modifications.

Comment 4: “Conclusions and Suggestions” should be changed to “Conclusions and Suggestions” or “Conclusions and Future outlook”.

Answer to Comment 4: Thank you for your valuable suggestion. We have already made modifications.

Finally, thank you again for your kind suggestions.

Sincerely,

Fajun Zhao  (on behalf of all the authors)